# Spindle architecture constrains karyotype evolution

Jana Helsen [1,2] ✉, Md Hashim Reza[3], Ricardo Carvalho[1], Gavin Sherlock [2] ✉ & Gautam Dey [1] ✉

The eukaryotic cell division machinery must rapidly and reproducibly duplicate and partition the cell's chromosomes in a carefully coordinated process. However, chromosome numbers vary dramatically between genomes, even on short evolutionary timescales. We sought to understand how the mitotic machinery senses and responds to karyotypic changes by using a series of budding yeast strains in which the native chromosomes have been successively fused. Using a combination of cell biological profiling, genetic engineering and experimental evolution, we show that chromosome fusions are well tolerated up until a critical point. Cells with fewer than five centromeres lack the necessary number of kinetochore-microtubule attachments needed to counter outward forces in the metaphase spindle, triggering the spindle assembly checkpoint and prolonging metaphase. Our findings demonstrate that spindle architecture is a constraining factor for karyotype evolution.

Chromosome fission, fusion and genome duplications are pervasive across the eukaryotic tree of life and can lead to dramatic differences in chromosome number, even between closely related species. A well-known example of rapid karyotype evolution is found in muntjac deer, whose number of chromosomes varies from $2n = 46$ in the Chinese muntjac *Muntiacus reevesi* to $2n = 6/7$ in the Indian muntjac *Muntiacus muntjak*[1]. The butterfly genus *Polyommatus* contains species with a haploid chromosome number ranging from $n = 10$ to $n = 226$ (ref. 2) and the ancestor of the model budding yeast, *Saccharomyces cerevisiae*, as a product of interspecies hybridization[3], effectively doubled its number of chromosomes from $n = 8$ to $n = 16$ overnight. Each of these examples highlights not only a case of dramatic karyotype rearrangement, but also shows that such changes can occur within relatively short evolutionary timeframes. Despite these changes, every chromosome must still be duplicated faithfully and segregated reliably during mitosis. Failure to do so results in aneuploidy, a state in which cells have an abnormal number of chromosomes, which leads to proteotoxic stress[4], and can result in certain birth defects[5] and cancers[6]. To allow for rapid karyotype evolution, the mitotic machinery must therefore be sufficiently robust to be able to support different genome configurations. Indeed, it is possible to fuse the 16 native chromosomes of the budding yeast *S. cerevisiae* into one single chromosome[7] or split them up into 33 smaller chromosomes[8] without killing the organism. However, it remains unclear whether such dramatic rearrangements still result in a stable interaction with the different structural components of the cell division machinery and whether this stability affects organismal fitness and the available trajectories for karyotype evolution. In this study, we use a combination of cell biological characterization and experimental evolution to determine the biophysical constraints dictating chromosome number evolution.

## Results

### Having fewer than five chromosomes causes a mitotic delay

To systematically explore how the cell division machinery copes with changes in chromosome number, we used a series of *S. cerevisiae* strains in which the 16 native chromosomes have been successively fused by concurrent telomere-to-telomere fusions and centromere excisions[7,9]. The resulting strains have chromosome numbers ranging from 16 all the way down to 1, with only minor changes in genome size and content (Fig. 1a). While budding yeast has been shown to tolerate these drastic

[1]Cell Biology and Biophysics, European Molecular Biology Laboratory, Heidelberg, Germany. [2]Department of Genetics, Stanford University School of Medicine, Stanford, CA, USA. [3]Molecular Mycology Laboratory, Molecular Biology and Genetics Unit, Jawaharlal Nehru Centre for Advanced Scientific Research, Bengaluru, India. ✉e-mail: jana.helsen@embl.de; gsherloc@stanford.edu; gautam.dey@embl.de

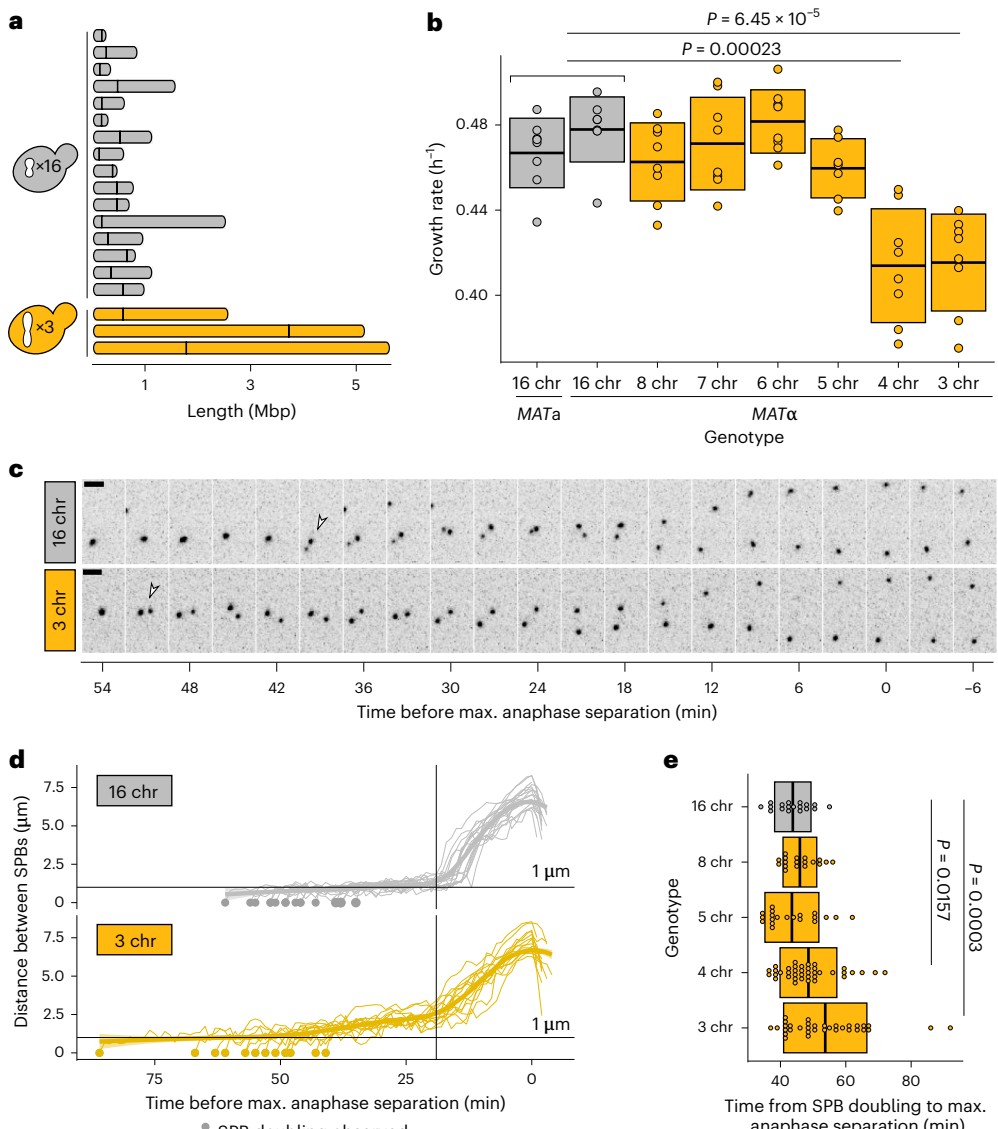

**Fig. 1 | Chromosome fusions induce spindle defects from 1n = 4.**
**a**, Chromosome lengths of wild-type (16 chr) and 3 chr *S. cerevisiae*. Vertical lines indicate positions of centromeres. **b**, Maximum growth rates of fused-chromosome strains on synthetic complete medium with 2% dextrose (SCD). Boxes show the means and s.d. Means were compared with the wild type using a two-tailed Student's *t*-test (from left to right, *n* = 8, 8, 8, 8, 8, 7, 8 and 8 independent population measurements). **c**, Montage of SPB (Spc42-mCherry) dynamics during mitosis for 16- and 3-chromosome strains. Scale bar, 2 μm;

intervals are 3 min. The time point with maximal SPB separation during anaphase was set to zero. Open arrows indicate SPB doubling. **d**, Distance between SPBs over time. For normalization, the time point with maximal SPB separation during anaphase was set to zero. The vertical line represents the inflection point (~start of anaphase). **e**, The time from SPB doubling to max. anaphase separation. Boxes represent the mean and s.d. Means were compared using a two-tailed Student's *t*-test (from left to right, *n* = 18, 18, 20, 39 and 34 independent single-cell measurements). chr, chromosome. Source numerical data are available in source data.

reductions in chromosome number[7,9], we find that such reductions also come at a fitness cost. By carefully measuring the growth rates of each strain in the series, we show that strains with fewer than five chromosomes have a growth defect corresponding to a 5–10-min increase in doubling time across experiments (Fig. 1b, Extended Data Fig. 1a and a summary of effect across multiple experiments in Fig. 4f; for reference, budding yeast doubles about every 1.5 h). Next, we wanted to test whether this growth defect can be explained by a delay in mitosis. To do so, we measured the distance between spindle pole bodies (SPBs; Spc42-mCherry) over time, from the moment of pole duplication to the end of anaphase (Fig. 1c,d and Extended Data Fig. 1b). We show that the time from SPB doubling to the end of anaphase is significantly longer in cells with a growth defect (Fig. 1e). Similarly to what we observe on a population level, single cells with three chromosomes take on average 8 min longer to progress through mitosis. As anaphase duration is

similar across genotypes (Fig. 1d and Extended Data Fig. 1b), we hypothesize that the mitotic delay is primarily due to a delay in metaphase. We also note that while the distance between SPBs during metaphase remains relatively stable at around 1 μm in wild-type cells, it steadily increases in strains with fewer chromosomes (Fig. 1d and Extended Data Fig. 1b). Strains with fewer chromosomes also exhibit increased spindle curvature (Extended Data Fig. 1c–e), as well as atypical distortions of the nuclear envelope during mitosis (Extended Data Fig. 1f). Together, these data indicate that the cell division machinery robustly tolerates chromosome fusions up until 1n = 5. However, strains with fewer chromosomes experience mitotic defects.

## Diploidization during laboratory evolution fixes the delay
Next, we sought to determine the molecular mechanisms that underlie the observed growth defect associated with low chromosome numbers.

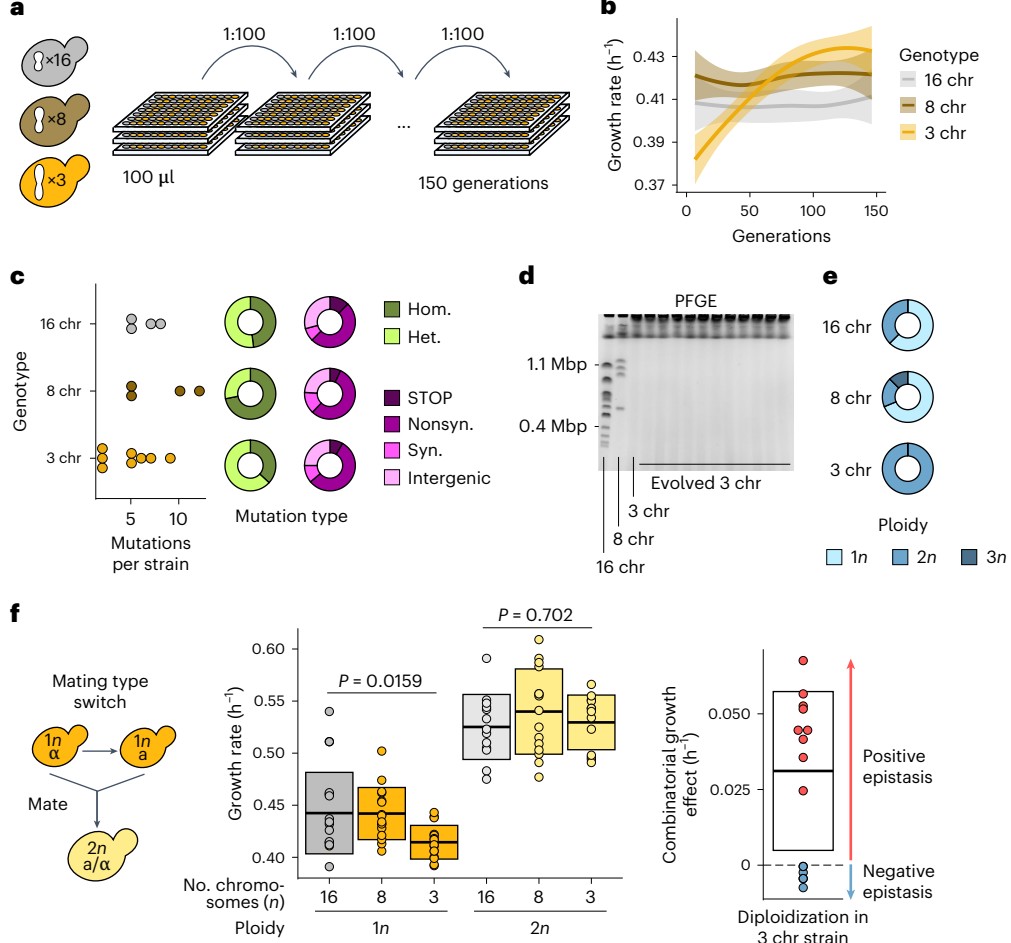

**Fig. 2 | Defects are overcome by diploidization during experimental evolution. a**, Schematic overview of the evolution experiments. Replicate populations of strains with either 16, 8 or 3 chromosomes were inoculated in 96-well plates filled with 100 µl SCD and 1:100 of each culture was transferred daily for a total of ~150 generations. **b**, Maximum growth rate on SCD over the course of evolution separated by genotype. Curves are smoothed and represent the average trend of eight replicate evolving populations. Ribbons represent 95% confidence intervals. **c**, Mutations observed in selected evolved strains. The number of mutations per sequenced strain is shown on the left, the proportions of homozygous (Hom.) and heterozygous (Het.) mutations are shown in green and the pink charts show the proportion of nonsense mutations and frameshifts (STOP), nonsynonymous (Nonsyn.) mutations, synonymous (Syn.) mutations and intergenic mutations. **d**, Representative pulsed-field gel electrophoresis

(PFGE) gel showing the karyotype of the three ancestral genotypes and 11 evolved three-chromosome strains. The remaining four gels can be found in Extended Data Fig. 2. **e**, The proportion of ploidies observed in evolved strains separated by genotype. For evolved 16- and 8-chromosome strains, clones from 16 populations were checked for ploidy. For evolved three-chromosome strains, all 56 populations were checked for ploidy. **f**, To make diploids, the mating type was switched using a plasmid with inducible HO endonuclease, and cells were allowed to mate to form diploids (left), maximum growth rates of haploid and diploid fused-chromosome strains (middle) and epistasis between chromosome number and ploidy (right). Boxes represent the means and s.d. Means were compared using a two-tailed Student's *t*-test (from left to right, $n = 16, 16, 15, 13, 15$ and $13$; independent population measurements). Source numerical data and unprocessed gels are available in source data.

By using experimental evolution, we can evaluate how specific defects can be repaired during evolution, offering insight into what caused the defect in the first place[10]. We established multiple replicate populations of strains with either 16, 8 or 3 chromosomes and evolved those populations in parallel for ~150 generations (Fig. 2a). We found that evolved strains with three chromosomes were able to completely overcome their initial fitness defect (Fig. 2b). Notably, these evolved strains acquired very few mutations (Fig. 2c), none of which was shared between independently evolved clones (Supplementary Table 1). Additionally, none of the strains evolved by chromosome fission (Fig. 2d and Extended Data Fig. 2). Instead, each and every population that was started from strains with three chromosomes adapted by auto-diploidization (Fig. 2e). Diploidization occurs frequently during laboratory evolution, primarily because diploid *S. cerevisiae* cells are known to be fitter than their haploid counterparts in conditions similar to the one that we used here[11]. However, the proportion of observed diploids is much higher for evolved 3-chromosome strains compared

with evolved 16- or 8-chromosome strains, suggesting that diploidization has a greater fitness benefit in strains with fewer chromosomes. To test whether there is positive epistasis between diploidization and low chromosome numbers, we generated isogenic diploids from the ancestral haploid strains and measured their growth rates (Fig. 2f). Although diploids are more fit overall as expected, the growth defect associated with low chromosome count disappears in three-chromosome diploids, showing that diploidization is sufficient to completely repair the growth defect.

## Five centromeres are sufficient to overcome the delay

Chromosomes in budding yeasts such as *S. cerevisiae* are each bound by just a single microtubule via its kinetochore[12,13], making it one of the simplest systems in which to study spindle dynamics. From a mitotic perspective, diploidization in this system therefore doubles the number of kinetochore microtubules (kMTs) within the cell. If the mitotic defect is caused by an insufficient number of kMTs or kMT

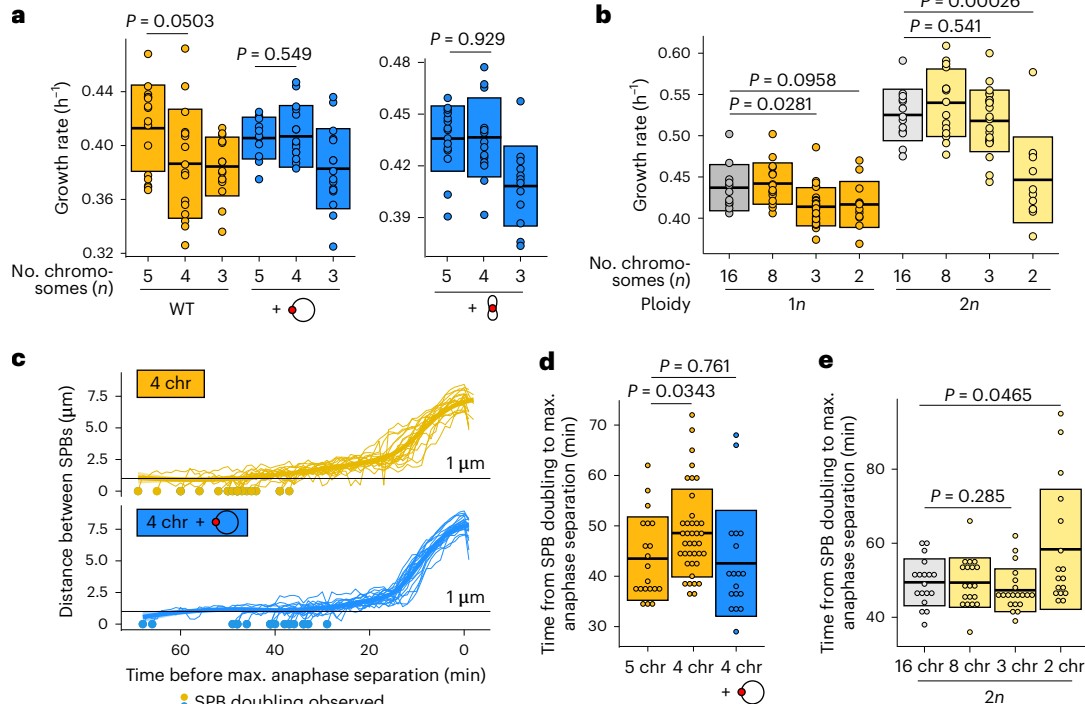

**Fig. 3 | Five centromeres are sufficient to overcome the mitotic defect.**
**a**, Maximum growth rates of fused-chromosome strains with and without additional centromere (red circle), supplied on either a plasmid or a small artificial chromosome. Boxes represent means and s.d. Means were compared using a two-tailed Student's *t*-test (from left to right, *n* = 16, 16, 16, 11, 15, 16, 15, 16 and 12 independent population measurements). **b**, Maximum growth rates of haploid and diploid fused-chromosome strains. Boxes represent means and s.d. Means were compared using a two-tailed Student's *t*-test (from left to right, *n* = 11, 16, 26, 12, 13, 15, 23 and 12 independent population measurements). **c**, Distance between SPBs over time for the four-chromosome strain with

and without centromeric plasmid. For normalization, the time point with maximal SPB separation during anaphase was set to zero. **d**, The time from SPB doubling to max. anaphase separation for the four-chromosome strain with and without centromeric plasmid. Boxes represent the means and s.d. Means were compared using a two-tailed Student's *t*-test (from left to right, *n* = 20, 39 and 18 independent single-cell measurements). **e**, The time from SPB doubling to max. anaphase separation for diploid strains. Boxes represent the means and s.d. Means were compared using a two-tailed Student's *t*-test (from left to right, *n* = 18, 18, 20 and 17 independent single-cell measurements). WT, wild type. Source numerical data are available in source data.

attachments (fewer than five), diploidization is an easy way for a strain with three chromosomes to increase the number above that threshold. One approach to test this hypothesis is to explore whether the defect can be fixed purely by increasing the total number of centromeres inside the cell. *S. cerevisiae* has a small 'point' centromere of ~120 bp, which can be easily put on a plasmid. Such centromeric plasmids have been shown to interact with the cell division machinery, and the number of kMTs has been shown to be directly proportional to the number of centromeric plasmids inside of a cell[14]. We introduced a centromeric plasmid into a strain with four chromosomes, and found that this indeed fixes both the growth defect (Fig. 3a) and the mitotic delay (Fig. 3c,d). In parallel, we transformed a small (9.7 kb) artificial chromosome into the same strains. This chromosome also carries a single centromere, but is more similar to native chromosomes because it is linear and has telomeres. We obtain the same result as for the centromeric plasmid: adding the artificial chromosome rescues the growth defect in the strain with four chromosomes, but not in the strain with three chromosomes (Fig. 3a). Together, this indicates that rather than the number or size of chromosomes, only the number of centromeres, and hence the number of kMTs and kMT attachments, determines whether a cell experiences a delay. Because adding a centromeric plasmid or artificial chromosome does not alter chromosome size or total chromosomal mass, these observations also preclude the possibility that the size of the fused chromosomes underlies the threshold; in other words, the possibility that the chromosomes would have become too large for efficient segregation. Likely, segregation of such large chromosomes is facilitated by increased chromosome condensation during mitosis[15]. This stands in contrast to what is known about mammalian systems, in

which chromosome size does seem to affect segregation efficiency[16]. If a budding yeast cell does indeed require more than four centromeres for stable growth, this would suggest that a diploid two-chromosome strain would still be below that limit and continue to show a growth defect. We diploidized the two-chromosome strain and measured growth rates and mitotic timing and find that this is indeed the case (Fig. 3b,e and Extended Data Fig. 3). Together, this shows that having five centromeres, regardless of the number of chromosomes or total DNA content, is sufficient to overcome both the growth and mitotic defect.

**Excess outward force in the metaphase spindle causes a delay**
kMT attachments are an important contributor to the force balance in metaphase spindles. During metaphase, motor proteins generate an outward force by pushing apart overlapping interpolar microtubules, and cohesion between sister chromatids generates an inward force through kMT attachments (Fig. 4a). Indeed, removing kinesin-5 motor proteins such as Cin8 or Kip1 shortens the metaphase spindle[14,17], whereas overexpression of *CIN8* lengthens it[18]. Additionally, reducing cohesion has been shown to elongate the metaphase spindle[14,19], whereas increasing the number of kMT attachments shortens it[14]. As noted above, we observe that the distance between SPBs during metaphase increases over time in cells with three chromosomes, much more so than in wild-type cells (Fig. 1d). By characterizing this phenotype in strains with differing numbers of chromosomes, we find that the extent of this phenomenon negatively correlates with the number of chromosomes (Fig. 4b and Extended Data Fig. 4a). This suggests that the net outward force in the metaphase spindle increases as the number of kMT attachments decreases, regardless of the total amount of DNA

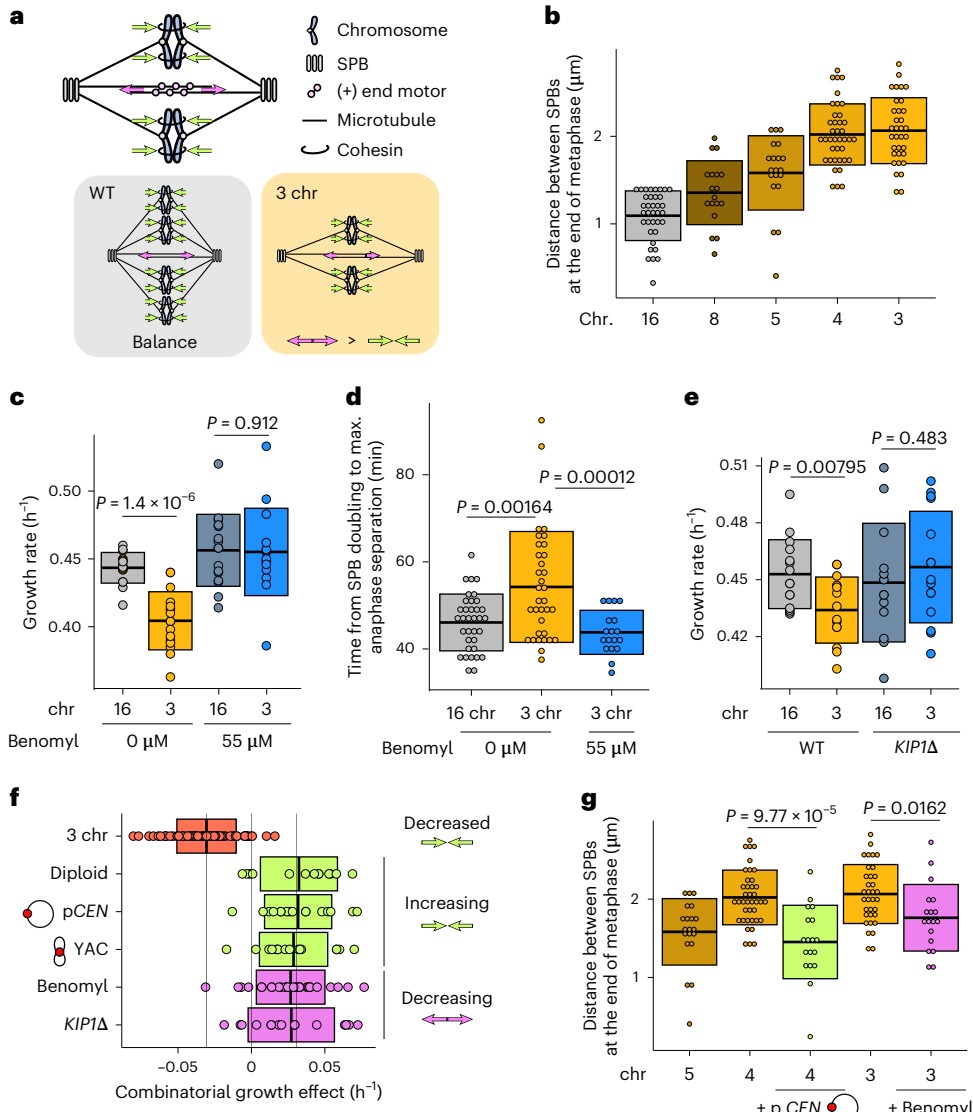

**Fig. 4 | Decreasing the net outward force in the mitotic spindle alleviates the defect. a**, Simplified schematic of inward (green arrows) and outward (pink arrows) forces in a metaphase spindle. **b**, Distance between SPBs at the end of metaphase for different fusion strains. **c**, Maximum growth rates of fused-chromosome strains with and without benomyl. Boxes represent means and s.d. Means were compared using a two-tailed Student's *t*-test (from left to right, *n* = 16, 16, 16 and 15 independent population measurements). **d**, The time from SPB doubling to max. anaphase separation. Boxes represent means and s.d. Means were compared using a two-tailed Student's *t*-test (from left to right, *n* = 35, 34 and 18 independent single-cell measurements). **e**, Maximum growth rates of fused-chromosome strains with and without *KIP1* deletion. Boxes represent

means and s.d. Means were compared using a two-tailed Student's *t*-test (from left to right, *n* = 15, 14, 13 and 15 independent population measurements). **f**, Summary of epistatic effects of different perturbations. Diploidization, adding a centromeric plasmid or adding an artificial chromosome increase the inward force and adding benomyl or deleting *KIP1* decrease the outward force. Boxes represent means and s.d. **g**, Distance between SPBs at the end of metaphase for different fusion strains and the effects of increasing inward force (+p*CEN*) or decreasing outward force (+benomyl). Boxes represent means and s.d. Means were compared using a two-tailed Student's *t*-test (from left to right, *n* = 20, 39, 18, 34 and 18 independent single-cell measurements). Source numerical data are available in source data.

inside the cell. The mitotic defect in strains with fewer than five kMT attachments could therefore be caused by excess outward force during metaphase. Both diploidization as well as the addition of centromeres would fix the defect by increasing the number of kMT attachments and as a result increasing the total amount of inward force. To test this hypothesis, we reduced the amount of outward force by treating the cells with a low concentration of benomyl. Benomyl is a tubulin-binding drug, which at low concentrations can decrease metaphase spindle length by suppressing microtubule dynamics without inducing detachments[20,21]. Treatment with benomyl did indeed fix both the growth and mitotic defect (Fig. 4c,d and Extended Data Fig. 4b). Additionally, we removed the motor protein Kip1 as an orthogonal approach for decreasing the outward force and this too rescued the growth defect (Fig. 4e).

In summary, the growth defect can be completely rescued by either increasing the inward force or decreasing the outward force in the metaphase spindle (Fig. 4f,g), which implies that an excess of outward force can only be tolerated up until a critical threshold.

### Kinetochore declustering and SAC involvement
As shown in Fig. 4b, the net outward force increases steadily as the number of chromosomes decreases, even in strains without a growth defect or mitotic delay. To explore how this excessive outward force causes an abrupt mitotic defect, we tested whether the spindle assembly checkpoint (SAC) is triggered in these cells. As long as the SAC remains active, transition from metaphase to anaphase is prevented. We deleted *MAD2*, a component of the SAC, which rescued the defect

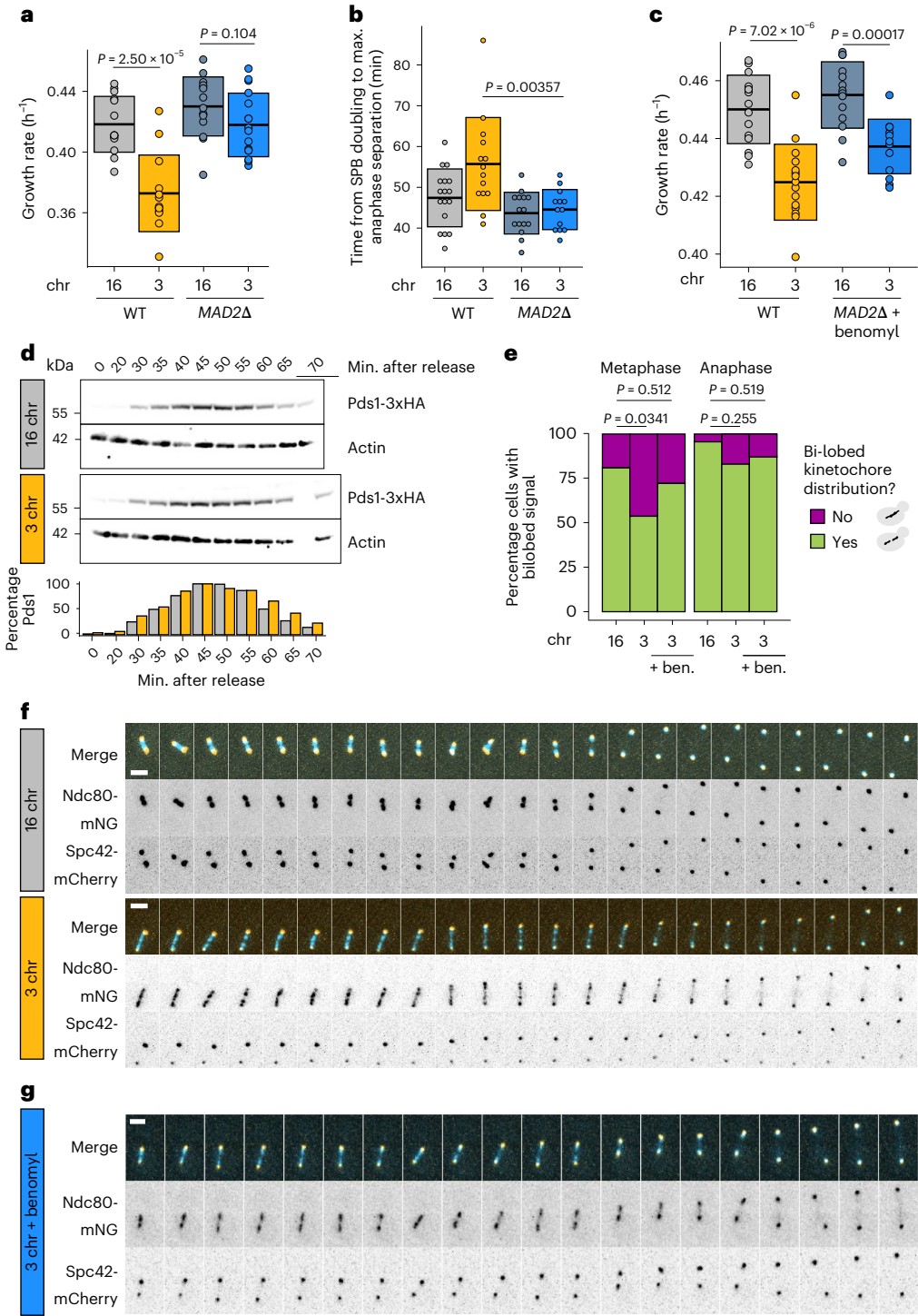

**Fig. 5 | The force imbalance causes kinetochore declustering and triggers the SAC. a**, Maximum growth rates of fused-chromosome strains with and without *MAD2* deletion. Boxes represent means and s.d. Means were compared using a two-tailed Student's *t*-test (from left to right, *n* = 14, 13, 15 and 16 independent population measurements). **b**, The time from SPB doubling to max. anaphase separation. Boxes represent the means and s.d. Means were compared using a two-tailed Student's *t*-test (from left to right, *n* = 17, 14, 15 and 13 independent single-cell measurements). **c**, Maximum growth rates of fused-chromosome strains with and without *MAD2* deletion + benomyl treatment (55 μM). Boxes represent means and s.d. Means were compared using a two-tailed Student's *t*-test (from left to right, *n* = 15, 15, 14 and 13 independent population measurements). **d**, Western blot analysis of Pds1 levels after G1 release for 16- and 3-chromosome strains. Cells were collected at the indicated time points and alpha-factor pheromone was added again 45 min after release to prevent the cells

from entering a second cell cycle. Actin levels were used as a loading control. Pds1-normalized values are shown in the bar plot at the bottom. An independent repeat of this experiment with analysis of the second cell cycle can be found in Extended Data Fig. 5f. **e**, Proportion of cells with a bi-lobed kinetochore signal (bi-modal Ndc80 signal along the spindle pole to spindle pole axis) in both metaphase and anaphase. Proportions were compared using a two-proportions *z*-test (from left to right, *n* = 26, 56, 100, 24, 41 and 31 independent single-cell measurements). **f**, Montage of kinetochore (Ndc80-mNG) and SPB (Spc42-mCherry) dynamics during metaphase for 16- and 3-chromosome strains. Scale bar, 2 μm; intervals are 30 s. **g**, Montage of kinetochore (Ndc80-mNG) and SPB (Spc42-mCherry) dynamics during metaphase in three-chromosome strains with benomyl (55 μM). Scale bar, 2 μm; intervals are 1 min. Source numerical data and unprocessed blots are available in source data.

(Fig. 5a,b and Extended Data Fig. 5a,e), suggesting that activation of the checkpoint underlies the delay in the three-chromosome strain. Deleting components of other cell cycle checkpoints, such as the DNA damage checkpoint or spindle positioning checkpoint, did not alleviate or exacerbate the defect (Extended Data Fig. 5b–d). For a more direct readout of mitotic timing and SAC activation, we measured Pds1 (securin) levels upon release from a G1 arrest. During metaphase, securin binds and inhibits separase, the protease that degrades cohesin. During the metaphase-to-anaphase transition, securin is degraded, releasing separase so that cohesin can be degraded and the sister chromatids separated[22]. SAC activation stabilizes Pds1 and therefore prevents this transition, making Pds1 levels a commonly used read-out for SAC activity[23]. Consistent with the 5–10-min delay we observe in the experiments shown above, the three-chromosome strain displays elevated Pds1 levels at the end of the cell cycle compared with the wild type (+15%), both during the first (Fig. 5d) and second cell cycle (Extended Data Fig. 5f) after G1 release. Together, these observations further support the conclusion that the mitotic delay is more precisely a metaphase delay, in line with our previous experiments. Next, we tested whether there is epistasis between force perturbations and the *MAD2* deletion. If force perturbations and inactivation of the checkpoint fixed the defect by affecting different cellular processes, their effect on growth rate would be additive. Instead, we see negative epistasis (Fig. 5c), indicating that there is a causal link between the force imbalance and the triggering of the SAC.

How does excess outward force in the metaphase spindle lead to SAC activation? During metaphase, sister chromatids must bi-orient at the centre of the mitotic spindle to ensure proper segregation during anaphase. The SAC prevents anaphase initiation until kinetochores from all sister chromatids are correctly attached to kinetochore microtubules from opposite poles. We tagged Ndc80, an outer kinetochore component, to visualize kinetochore dynamics during metaphase and observed that kinetochores fail to properly cluster during metaphase in strains with the growth defect (Fig. 5e,f), but re-establish clustering during anaphase. Reducing the amount of outward force by treating the cells with a low concentration of benomyl improves clustering; although the kinetochore foci are still not as defined as in wild-type cells (Fig. 5g), the kinetochore distribution along the spindle during metaphase is restored to a more bi-lobed (bi-modal) distribution (Fig. 5e).

While further research will be necessary to unravel the exact molecular underpinnings of how excess outward force in the metaphase spindle ultimately leads to activation of the SAC, one hypothesis is that the excess force causes the metaphase spindle to elongate too fast to allow for efficient sister kinetochore pairing, leading to low tension at the kinetochores. Low tension is a signal for improper bi-orientation and leads to microtubule detachment through activation of the Aurora B-dependent error correction mechanism[24]. Detached kinetochores in turn trigger the SAC[25]. Alternatively, if the mitotic defect is not caused by tension-dependent detachment, our observations could also be explained by deregulation of kMT length. In *S. cerevisiae*, the length of kMTs has been proposed to control discrimination of bi-oriented from syntelic attachments during metaphase[26]. In this scenario, declustered kinetochores could still be attached to kMTs, but the kMTs might be too long for efficient detection of bi-orientation.

## Discussion
Our results show that the spindle architecture of budding yeast robustly supports karyotypes with at least five chromosomes. Below that, cells experience reduced fitness. In nature, the lowest chromosome number observed in other yeast species with similar simple point centromeres is six, in *Kluyveromyces lactis*, a haploid species with a similar genome size to *S. cerevisiae*[27,28]. Budding yeasts have a small spindle with just a single kMT per chromosome, which may exacerbate the effect of low chromosome count on mitosis. Indeed, fission yeast (*Schizosaccharomyces pombe*), which has larger regional centromeres with

two to four kMT attachments each[29], has no problem segregating its three native chromosomes. Regardless, budding yeasts are not the only eukaryotes with small spindles. *Ostreococcus tauri*, a species of marine green algae, has a spindle composed of only ten microtubules[30] and it would be interesting to see whether our model can be applied to determine this and other clades' evolutionary limitations on karyotype. Even in species with larger spindles, there is evidence that dramatic karyotypic changes can put evolutionary pressure on components of the cell division machinery. The Indian muntjac *M. muntjak*, whose chromosome number was reduced to $2n = 6/7$ through an estimated 26 lineage-specific chromosome fusion events[1], has centromeres that are much larger than those found in its sister species *M. reevesi* ($2n = 46$)[31]. Notably, these large centromeres can bind up to 60 kMTs[32]. Additionally, the genome of *M. muntjac* shows signatures of positive selection in kinetochore proteins CENP-Q and CENP-V[1]. In *Cochlearia*, a plant genus comprising diploid, tetraploid and hexaploid species, changes in ploidy were shown to correlate with evolution in several kinetochore components, including CENP-E, CENP-C and INCENP[33]. Our work shows that karyotype and the cell division machinery are inherently linked during evolution and it provides insight into how the mechanics of a core cellular process can constrain evolutionary trajectories.

## Online content

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

## Methods

### Strains

Full genotypes of all strains used in this study can be found in Supplementary Table 2. Strains were constructed using the standard LiAc-based transformation protocol for budding yeast. Proteins were tagged C-terminally with mNeonGreen using a mNeonGreen:HphNT1 cassette, with mCherry using a mCherry:NatNT2 cassette, with 3xHA using a 3xHA:HphNT1 cassette and Tub1 was tagged N-terminally using Addgene vector #50654 (ref. [34]). Genes were deleted using a KanMX4 cassette. Strains were diploidized by mating-type switching and mating in batch, using a plasmid containing HO endonuclease[35] and ploidy was verified using PI staining (see below). Mating types were confirmed using mating-type-specific primers[36]. Centromere plasmids contained *CEN4*, *ARS1* and either KanMX4 or HphMX for selection. Cloning vector pYAC4 was used as a small artificial chromosome after linearization[37]. All oligonucleotides used for strain construction are listed in Supplementary Table 3.

### Growth assays

All growth assays were performed on synthetic complete medium with 2% dextrose (SCD). Per genotype, two biological replicates were inoculated in 150 μl SCD and serially diluted for overnight growth at 30 °C in a 96-well plate. The next morning, log-phase cultures at $OD_{600}$ < 0.1 were selected and 5 μl of each selected culture was inoculated in 95 μl SCD in a 96-well plate. For each strain, this was repeated so that there were 16 replicate measurements per genotype in total. By consecutive culturing, we ensured that every culture is growing exponentially before the start of the growth experiment. The plate was incubated at 30 °C with continuous double orbital shaking in a BioTek Epoch2 microplate reader (Agilent) using Gen5 software (v.3.11, BioTek) and the $OD_{600}$ was measured every 10 min for 24 h. Maximum growth rates were determined using the gcplyr package in R[38]. For growth assays with benomyl (Sigma) or hydroxyurea (Sigma), all strains were pre-grown on SCD without the chemical compound and the chemical compound was added for the actual growth assay.

### Live-cell microscopy

Strains were imaged live in eight-chamber glass bottom dishes (ibidi) in SCD at 30 °C. The dishes were coated with 1 mg ml⁻¹ concanavalin A (Sigma) and log-phase cells were pipetted into the chambers, after which they were allowed to settle for 30 min at 30 °C. The medium was removed by pipetting and replaced by fresh SCD after one wash to remove unattached cells. All live-cell microscopy experiments were conducted using an Olympus IXplore SpinSR spinning-disc confocal microscope with CSU-W1 (Yokogawa), 50-μm pinholes and a Flash4 sCMOS camera (Hamamatsu). Samples were illuminated with 488 nm (mNeonGreen or CloverGFP) and 561 nm (mCherry) lasers. The microscope was controlled by cellSens Dimension software (v.3.2, Olympus). SPB time-lapses, spindle time-lapses and spindle snapshots were made using a UPLXAPO ×60 oil immersion objective (NA 1.42, Olympus). Kinetochore time-lapses and snapshots were made using a UPLSAPO-S ×100 silicone immersion objective (NA 1.35, Olympus). For spindle time-lapses, cells were imaged using 15 z-stacks with a step size of 0.27 μm, for spindle snapshots, 23 z-stacks with a step size of 0.27 μm, for SPB time-lapses, 15 z-stacks with a step size of 0.36 μm, for kinetochore time-lapses, 15 z-stacks with a step size of 0.28 μm and for kinetochore snapshots, 23 z-stacks with a step size of 0.28 μm.

### Image analysis

Fiji[39] (ImageJ2 v.2.9.0) was used for basic image processing (cropping, z-stack projections, scaling and look up table (LUT) selection) and for measuring spindle curvature and SPB distance over time. To measure SPB distance over time, maximum intensity projections were made of 3-h time-lapses of Spc42-tagged strains, using 1-min intervals. Per genotype, ~20 regions of interest (ROIs) were selected of cells in which SPBs could be followed from the moment of duplication up until collapse of the spindle. The straight-line tool was used to measure inter-SPB distance for each time point. To quantify spindle curvature, maximum intensity projections were made of snapshot images of Tub1-tagged strains. Within each image, up to ten ROIs were selected of cells with clear anaphase spindles. For each genotype, 50 ROIs were selected in total (5–6 different images). The straight-line tool was used to measure the distance between both ends of the spindle and the freehand-line tool was used to trace the spindle and estimate spindle length. The spindle curvature was defined as the difference between the two measurements, divided by the straight distance. To quantify the proportion of cells with bi-lobed kinetochore distributions, ROIs were selected of dividing cells with a clear Ndc80-mNG and Spc42-mCherry signal (wild type, n = 50; three-chromosome strain, n = 97; 3 chr + benomyl, n = 131). Using the straight-line tool, inter-SPB distance was measured and used to classify cells into metaphase (inter-SPB distance <1.25 μm for wild type and <2.5 μm for three-chromosome strain) or anaphase (inter-SPB distance >1.25 μm for wild type and >2.5 μm for three-chromosome strain). The rotated rectangle tool was used to select an area from SPB to SPB, after which a profile plot was generated to visualize the distribution of the Ndc80 signal. This plot in turn was used to quantify the proportion of cells with a bi-lobed (bi-modal) signal.

### Experimental evolution

For each genotype (16-, 8- and 3-chromosome strains), 56 replicate populations were established by inoculating single colonies in different wells of a 96-well plate containing 100 μl SCD. Populations were transferred daily around the same time, by inoculating 1 μl of old culture into 100 μl fresh SCD. Cells were grown at 30 °C with shaking. Every population reached saturation after 24 h, so we used the dilution factor (1:100) to estimate the number of generations per transfer (~6.7). To monitor average growth rate throughout evolution, one of the 96-well plates was evolved in the BioTek Epoch2 microplate reader (Agilent). Populations were frozen every fourth transfer and at time points of particular interest (for example, 100 generations). The experiment was stopped at 150 generations, at which point the growth rate data indicated that three-chromosome strains had repaired their growth defect. For sequencing and ploidy determination, frozen populations were streaked on YPD plates to isolate single clones. Whole populations were grown for pulsed-field gel electrophoresis.

### Whole-genome sequencing

The YeaStar genomic DNA kit (Zymo research) was used to isolate genomic DNA from single clones. Sequencing libraries were prepared using the Nextera kit as described previously[40], starting with 5–10 ng genomic DNA. The quality of the pooled libraries was assessed by measuring concentrations on the Qubit (Invitrogen) and fragment size distribution on a Bioanalyzer platform (Agilent). Samples were sent for paired-end sequencing on an Illumina HiSeq X, with an average read length of 150 bp. The quality of the reads was assessed using FastQC v.0.11.9 (Babraham Bioinformatics) and Nextera transposase sequences were trimmed using Trim galore! v.0.6.7 (Babraham Bioinformatics). Trimmed reads were mapped to the reference S288c genome (v.R64; NCBI, GCF_000146045.2) using bwa-mem v.0.7.17 with default settings[41]. Indels and single-nucleotide variants (SNVs) were called using GATK v.4.2.6.1 (ref. [42]) using HaplotypeCaller and default settings. Variants present in the ancestral strains were filtered out, as well as SNVs with a quality score below 175 and indels with a quality score below 200. Finally, all remaining SNVs and indels were verified by manual curation in IGV[43] v.2.12.3.

### Pulsed-field gel electrophoresis

Yeast chromosome plugs were prepared as described in the Bio-Rad CHEF-DR-III manual. In brief, 0.25 ml stationary overnight culture was

washed twice in 10 ml ice-cold 50 mM EDTA. Cells were resuspended in 250 µl cell suspension buffer (10 mM Tris, 50 mM EDTA and 2 mM NaCl), spun down, and resuspended in 40 µl cell suspension buffer. Then, 10 µl lyticase (Sigma) stock (1000 U ml$^{-1}$) was added and the cell suspension was mixed with 50 µl molten 2% CleanCut agarose (Bio-Rad) after which the mixture was pipetted into a plug mould (Bio-Rad). Plugs were allowed to solidify on ice and were then pushed out of the moulds into microcentrifuge tubes with 0.5 ml lyticase buffer (10 mM Tris and 50 mM EDTA) with 30 µl lyticase stock. The plugs were incubated at 37 °C for 2 h, after which they were transferred to new microcentrifuge tubes with 0.75 ml proteinase K buffer (10 mM Tris, 100 mM EDTA and 0.5% SDS) with proteinase K (23 U ml$^{-1}$, VWR). The plugs were incubated at 50 °C overnight, after which they were equilibrated in new microcentrifuge tubes with 0.5× TBE before insertion into the gel (half a plug per lane). The gel was made using Pulsed-Field-Certified Agarose (1%, Bio-Rad) in 0.5× TBE, and was run on a CHEF-DR-III Pulsed-Field Gel Electrophoresis System (Bio-Rad), at 6 V cm$^{-1}$, 120°, switch time 60–120 s, for 24 h at 14 °C. The gel was stained with GelRed (Sigma) for visualization.

### Ploidy determination

Ploidy was determined as described previously[44] by staining the cells with propidium iodide[44]. In brief, cells were fixed for 1 h in 70% ethanol, washed three times in 50 mM sodium citrate buffer, treated with 0.5 mg ml$^{-1}$ RNase A (NEB) at 37 °C for 2 h and stained overnight at 4 °C with 25 µg ml$^{-1}$ propidium iodide (Sigma). A haploid (BY4741) and diploid (BY4743) strain were used as controls and fluorescence of 30,000 cells was analysed by flow cytometry on an Acea Novocyte Quanteon (Agilent).

### Ultrastructure expansion microscopy

Ultrastructure expansion microscopy (U-ExM) was performed as described previously[45,46] with a few modifications. In brief, log-phase cells were fixed with 4% HCHO (FA) in PEM buffer (100 mM PIPES, 1 mM EGTA and 1 mM MgSO$_4$, pH 9.0), washed twice with 1× PBS and once with PEM-S (1.2 M sorbitol in PEM). The fixed cells were resuspended in PEM-S buffer and were enzymatically digested with 2.5 mg ml$^{-1}$ Zymolyase 20T at 37 °C with agitation for 15 min. Cells were washed once with PEM-S buffer. This was followed by overnight anchoring in acrylamide/formaldehyde (1% acrylamide and 0.7% formaldehyde diluted in 1× PBS) at 37 °C. The anchored cells were then allowed to attach to a 6-mm poly-L-lysine-coated coverslip for 1 h. Gelation was performed on ice using a monomer solution (19% ($wt/v$) sodium acrylate, 10% ($v/v$) acrylamide, 0.1% ($v/v$) N,N′-methylenebisacrylamide in PBS) and the gel was kept for polymerization for 1 h at 37 °C in a moist chamber. For denaturation, the gel was transferred to denaturation buffer (50 mM Tris, pH 9.0, 200 mM NaCl and 200 mM SDS, pH 9.0) and incubated at 95 °C for 1.5 h. Following denaturation, the gel was expanded with three subsequent washes with water. After expansion, the gel diameter was measured to determine the expansion factor. For U-ExM images, scale bars have not been rescaled for the gel expansion factor. Pan-labelling for U-ExM was carried out at 1:500 dilution with DyLight 594 NHS ester (Thermo Fisher Scientific, 46412) in 1× PBS overnight at 4 °C. For tubulin immunostaining, the gel was stained using YL1/2 anti-α-tubulin (rat) (a kind gift from G. Pereira, COS Heidelberg, Germany), as the primary antibody at 1:25 dilution and incubated overnight at 4 °C. The gel was then incubated with goat anti-mouse-IgG coupled to Alexa Fluor 488 (Invitrogen A11029) secondary antibody at 1:1,000 dilution and incubated at 37 °C for 3 h in the dark. The antibody dilutions were prepared in 3% BSA in 1× PBS with 0.1% Tween 20. The gel was washed thrice with PBS with 0.1% Tween 20 for 30 min at room temperature. The gel was expanded with three subsequent washes with water before imaging. For microscopy, poly-L-lysine-coated two-chamber glass-bottom dishes (ibidi) were used. Gels were cut to an appropriate size to fit the ibidi chambers. The gels were overlaid with water to prevent drying or

any shrinkage during imaging. The gels were imaged using the Zeiss LSM980 Airyfast confocal microscope using a Plan-Apochromat ×63/1.4 oil DIC M27 objective.

### Western blot for Pds1 dynamics after alpha-factor arrest

To be able to arrest the three-chromosome strain with alpha-factor, we switched the strain's mating type from *MAT*α to *MAT*a using the method described above under 'Strains'. Additionally, we endogenously tagged Pds1 with 3xHA in both the wild-type and three-chromosome strain. Cells were inoculated in 5 ml YPAD medium in the morning and this preculture was used to inoculate a 100 ml YPAD overnight culture so that the culture would reach OD$_{600}$ 0.2 the next morning. Cells were grown at 30 °C throughout the experiment. The next morning, cells were diluted once more and grown for an additional 1.5 h, so that we had stable 100 ml log-phase cultures of OD$_{600}$ 0.2 to start the arrest. The cells were then arrested in G1 using a final concentration of 4.0 µg ml$^{-1}$ alpha-factor mating pheromone (Zymo Research) and incubated at 30 °C with shaking until the vast majority of cells in the population exhibited the 'shmoo' phenotype (2.5 h). G1-arrested cells were released by washing them three times in pre-warmed YPAD after which they were grown at 30 °C and samples were collected as needed. For one of the two experiments we prevented the cells from going into a second cell cycle by adding 4.0 µg ml$^{-1}$ alpha-factor mating pheromone 45 min after release. For each time point, 2 ml of the culture was pelleted and snap-frozen using liquid nitrogen. The cell pellets were lysed by TCA precipitation[47] and resuspended in 50 µl High Urea dithiothreitol (DTT) buffer (200 mM Tris-HCl, pH 6.8, 8 M urea, 5% $w/v$ SDS, 1 mM EDTA, 100 mM DTT and bromophenol blue). Pds1 levels were monitored using a mouse anti-HA antibody (cat. no. 26183, Invitrogen, 1:1,000 dilution), actin levels were analysed using a mouse anti-actin antibody (cat. no. MAB1501R, Chemicon, 1:1,000 dilution) and an HRP-conjugated goat anti-mouse secondary antibody (cat. no. 31430, Invitrogen, 1:10,000 dilution). The blots were developed using a chemiluminescence substrate (Millipore cat. no. WBULP) and imaged using an Azure 280 imaging system (Azure). The Pds1 values shown in the plots were normalized to the time point with the highest Pds1 signal (45 min for the experiment shown in Fig. 5d, 105 min for the experiment shown in Extended Data Fig. 5f).

### Statistics and reproducibility

Sample sizes were chosen to be comparable with published manuscripts within the field[48,49]. For population growth rate measurements and single-cell microscopy experiments, experiments were included if the control strains that were included in each experiment behaved as expected. Population growth experiments were each set up with 16 independent replicates and only replicates for which the gcplyr package failed to calculate a growth rate (for example due to bubble formation during incubation) were removed. Exact $n$ values for each experiment can be found in the figure legends. For single-cell microscopy, 20 independent cells were chosen at random for each SPB tracking experiment and only cells for which the SPB signal could not be reliably identified for three or more sequential frames were removed from further analysis. Exact $n$ values for each experiment can be found in the figure legends. Fifty cells were chosen for spindle curvature measurements. A total of 56 independent replicate populations were established for each genotype in the evolution experiments, of which one clone per population was sequenced for 4 (16-chromosome strain), 4 (8-chromosome strain) and 8 (3-chromosome strain) populations and 16 (16-chromosome strain), 16 (8-chromosome strain) and 56 (3-chromosome strain) were checked for ploidy. No further data were excluded from the analyses. All raw measurements can be found in the source data. Western blots were performed in duplicate using independent experiments. The experiments were not randomized. The investigators were not blinded to the allocation during experiments and outcome assessment.

**Reporting summary**

Further information on research design is available in the Nature Portfolio Reporting Summary linked to this article.

## Data availability

All images used for data analysis are available on figshare at https://doi.org/10.6084/m9.figshare.c.6890251.v2 (ref. 50). Sequencing data were deposited on ENA under accession number PRJEB67700. Raw measurements used for plotting can be found in source data. All other data supporting the findings of this study are available from the corresponding author on reasonable request. Source data are provided with this paper.

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

## Acknowledgements

The authors thank O. Dudin, M. Hays, B. Baum, J. van Gestel, L. Steinmetz, I. Raote, F. Mikus and other members of the Dey and Sherlock laboratories for critical feedback on the manuscript. Funding was provided by Life Science Alliance Bridging Excellence Fellowship (J.H.); EMBL Corporate Partnership Programme Fellowship (H.R.); EMBO Scientific Exchange Grant 10212 (H.R.); Department of Biotechnology-Research Associate Fellowship (H.R.); NIGMS R35 GM131824 (G.S.); European Molecular Biology Laboratory (J.H., R.C. and G.D.); and the European Union (ERC, KaryodynEvo, 101078291) (J.H. and G.D.).

## Author contributions

Conceptualization: J.H., G.S. and G.D. Methodology: J.H., G.S. and G.D. Investigation: J.H., H.R., R.C., G.S. and G.D. Formal analysis: J.H. Visualization: J.H. Funding acquisition: J.H., G.S. and G.D. Project administration: J.H., G.S. and G.D. Supervision: G.S. and G.D. Writing - original draft: J.H., H.R., G.S. and G.D. Writing - review & editing: J.H., H.R., G.S. and G.D.

## Funding

## Competing interests

The authors declare no competing interests.

## Additional information

**Extended data** is available for this paper at https://doi.org/10.1038/s41556-024-01485-w.

**Correspondence and requests for materials** should be addressed to Jana Helsen, Gavin Sherlock or Gautam Dey.

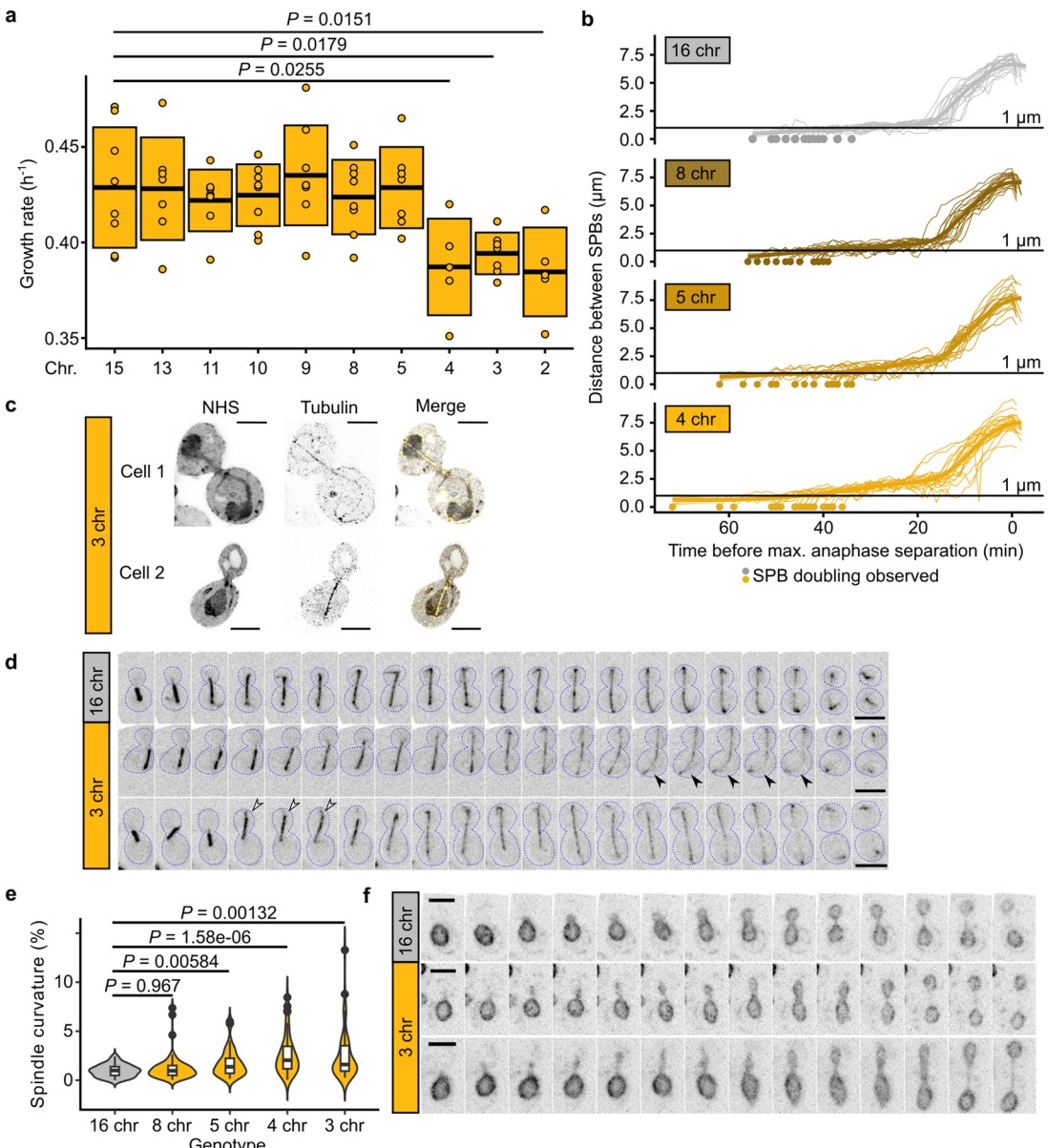

**Extended Data Fig. 1 | Growth and mitotic defects in fused-chromosome strains. (a)** Maximum growth rates of fused-chromosome strains on synthetic complete medium with 2% dextrose (SCD). Boxes show the means and standard deviation. Means were compared using a two-tailed Student's *t*-test (from left to right, *n* = 8, 7, 7, 8, 8, 8, 7, 5, 7, 5; independent population measurements). **(b)** Distance between SPBs over time for different fusion strains. For normalization, the time point with maximal SPB separation during anaphase was set to zero. **(c)** Expanded cells with spindle defects. Cells were labelled with pan protein label NHS ester and for tubulin. Scale bar = 10 µm, expansion factor = 4.18. **(d)** Montage of spindle dynamics over time (CloverGFP-tub1). Scale bar = 5 µm, intervals are

1 min. Closed arrows point to an example of increased spindle curvature, open arrows to an example of the whole spindle moving into the daughter cell. **(e)** Spindle curvature (%), calculated as the total spindle length relative to the distance between spindle pole bodies (SPBs). Data are represented as violin plots with boxes representing the interquartile range, the lines representing the median and the whiskers indicating the minimum and maximum values. Black dots represent outliers. *n* = 50 for each genotype; independent single-cell measurements. Distributions were compared using Kolmogorov-Smirnov tests. **(f)** Montage of nuclear envelope (Hmg1-mCherry) dynamics over time. Scale bar = 5 µm, intervals are 1 min. Source numerical data are available in source data.

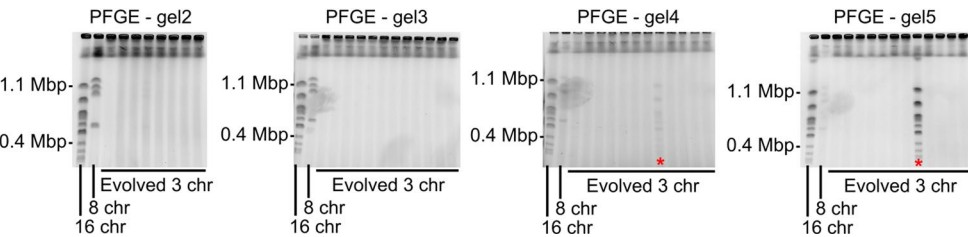

**Extended Data Fig. 2 | PFGE of evolved strains shows no chromosome fission.** PFGE gels showing the karyotype of the evolved 3-chromosome populations. Red stars indicate populations with cross-contamination of a wild-type strain. Chromosome numbers of clones isolated from these populations were double-checked before ploidy determination and sequencing. Unprocessed gels are available in source data.

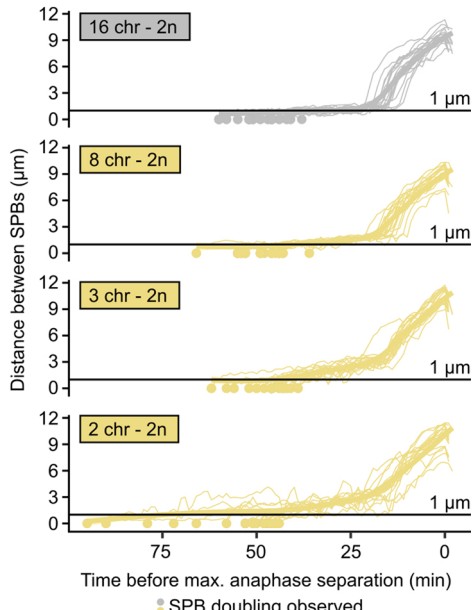

**Extended Data Fig. 3 | Distance between SPBs over time in diploids.** For normalization, the time point with maximal SPB separation during anaphase was set to zero. Source numerical data are available in source data.

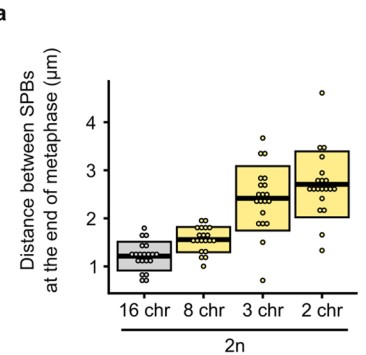

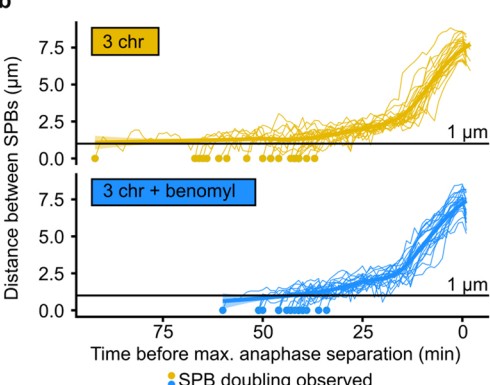

**Extended Data Fig. 4 | Distance between SPBs after changing the net outward force in the metaphase spindle.** **(a)** Distance between SPBs at the end of metaphase for different diploid fusion strains. Boxes represent means and standard deviation (from left to right, $n = 18, 18, 20, 17$; independent single-cell measurements). **(b)** Distance between SPBs over time in benomyl-treated cells. For normalization, the time point with maximal SPB separation during anaphase was set to zero (top: $n = 20$, bottom: $n = 18$). Source numerical data are available in source data.

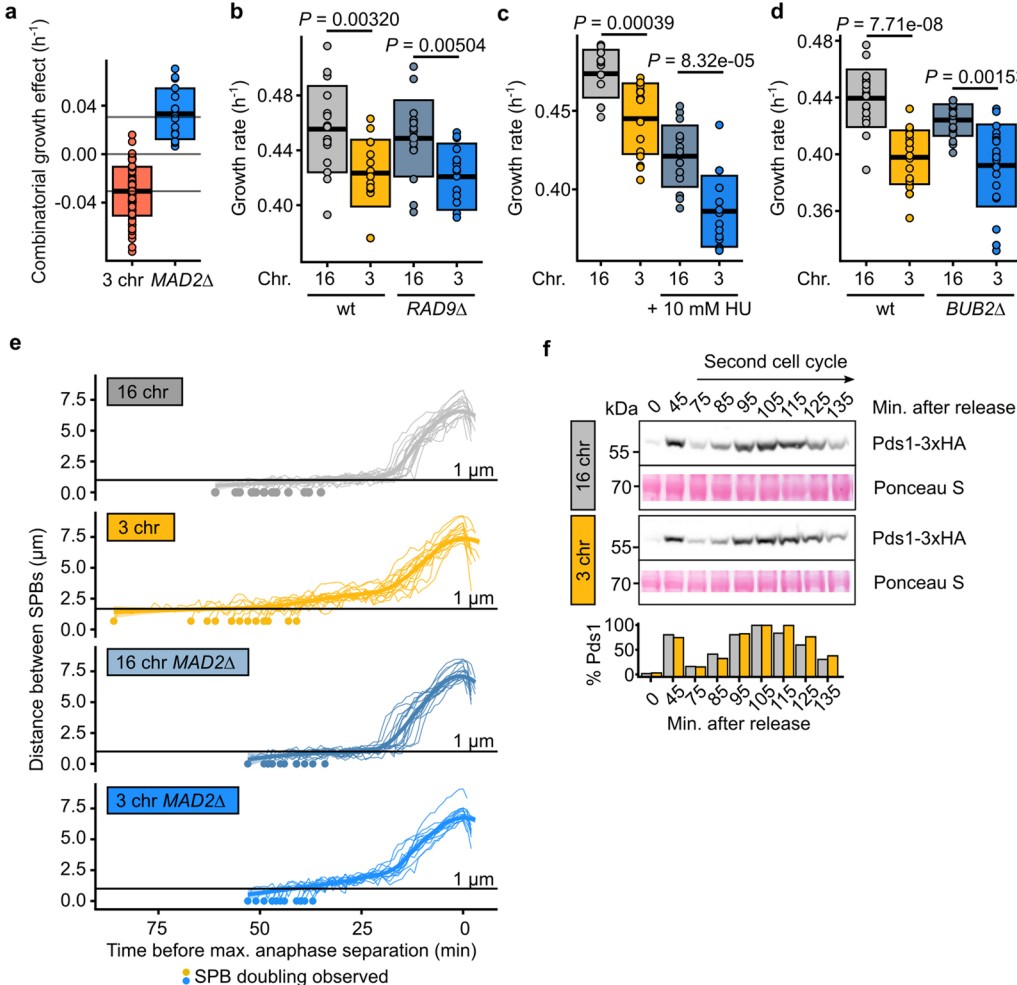

**Extended Data Fig. 5 | SAC activation in fused-chromosome strains.**
**(a)** Epistatic effect of *MAD2* deletion. Boxes represent means and standard deviation. **(b)** Maximum growth rates of fused-chromosome strains with and without *RAD9* deletion. Boxes represent means and standard deviations. Means were compared using a two-tailed Student's *t*-test (from left to right, $n$ = 16, 16, 16, 16; independent population measurements). **(c)** Maximum growth rates of fused-chromosome strains with and without 10 mM hydroxyurea (HU). Boxes represent means and standard deviations. Means were compared using a two-tailed Student's *t*-test (from left to right, $n$ = 13, 16, 16, 15; independent population measurements). **(d)** Maximum growth rates of fused-chromosome strains with and without *BUB2* deletion. Boxes represent means and standard deviations.

Means were compared using a two-tailed Student's *t*-test (from left to right, $n$ = 19, 21, 21, 21; independent population measurements). **(e)** Distance between SPBs over time in *MAD2Δ* cells. For normalization, the time point with maximal SPB separation during anaphase was set to zero. **(f)** Western blot analysis of Pds1 levels after G1 release for 16- and 3-chromosome strains, focused on the second cell cycle after release. Cells were collected at the indicated time points. Ponceau S staining was used as a loading control. Pds1-normalized values are shown in the bar plot at the bottom. An independent repeat of this experiment with analysis of the first cell cycle can be found in Fig. 5d. Source numerical data and unprocessed blots are available in source data.

Gavin Sherlock
Jana Helsen

# Reporting Summary

## Statistics

For all statistical analyses, confirm that the following items are present in the figure legend, table legend, main text, or Methods section.

| n/a | Confirmed | |
|---|---|---|
| ☐ | ☒ | The exact sample size (*n*) for each experimental group/condition, given as a discrete number and unit of measurement |
| ☐ | ☒ | A statement on whether measurements were taken from distinct samples or whether the same sample was measured repeatedly |
| ☐ | ☒ | The statistical test(s) used AND whether they are one- or two-sided<br>*Only common tests should be described solely by name; describe more complex techniques in the Methods section.* |
| ☐ | ☒ | A description of all covariates tested |
| ☐ | ☒ | A description of any assumptions or corrections, such as tests of normality and adjustment for multiple comparisons |
| ☐ | ☒ | A full description of the statistical parameters including central tendency (e.g. means) or other basic estimates (e.g. regression coefficient) AND variation (e.g. standard deviation) or associated estimates of uncertainty (e.g. confidence intervals) |
| ☐ | ☒ | For null hypothesis testing, the test statistic (e.g. *F*, *t*, *r*) with confidence intervals, effect sizes, degrees of freedom and *P* value noted<br>*Give P values as exact values whenever suitable.* |
| ☒ | ☐ | For Bayesian analysis, information on the choice of priors and Markov chain Monte Carlo settings |
| ☒ | ☐ | For hierarchical and complex designs, identification of the appropriate level for tests and full reporting of outcomes |
| ☒ | ☐ | Estimates of effect sizes (e.g. Cohen's *d*, Pearson's *r*), indicating how they were calculated |

*Our web collection on statistics for biologists contains articles on many of the points above.*

## Software and code

Policy information about availability of computer code

| Data collection | Gen5 software (version 3.11, BioTek) was used to control the BioTek Epoch2 microplate readers (Agilent) during growth rate measurements. For live-cell microscopy the microscope was controlled by cellSens Dimension software (version 3.2, Olympus). |
|---|---|
| Data analysis | Maximum growth rates were determined using the gcplyr package in R (version 1.5.2). Fiji (ImageJ2 version 2.9.0)  was used for basic image processing (cropping, z-stack projections, scaling, LUT selection), and for measuring spindle curvature and SPB distance over time. For analysis of sequencing data, the quality of the reads was assessed using FastQC version 0.11.9 (Babraham Bioinformatics), and Nextera transposase sequences were trimmed using Trim galore! version 0.6.7 (Babraham Bioinformatics). Trimmed reads were mapped to the reference S288c genome (version R64) using bwa-mem version 0.7.17 with default settings. Indels and SNVs were called using GATK version 4.2.6.1, using HaplotypeCaller and default settings. High-quality SNVs and indels were verified by manual curation in IGV version 2.12.3. |

For manuscripts utilizing custom algorithms or software that are central to the research but not yet described in published literature, software must be made available to editors and reviewers. We strongly encourage code deposition in a community repository (e.g. GitHub). See the Nature Portfolio guidelines for submitting code & software for further information.

## Data

Policy information about availability of data

All manuscripts must include a data availability statement. This statement should provide the following information, where applicable:

- Accession codes, unique identifiers, or web links for publicly available datasets
- A description of any restrictions on data availability
- For clinical datasets or third party data, please ensure that the statement adheres to our policy

All images used for data analysis are available at doi.org/10.6084/m9.figshare.c.6890251.v2. Sequencing data were deposited on ENA under accession number PRJEB67700. Raw measurements used for plotting can be found in Source Data. Source data have been provided in Source Data. All other data supporting the findings of this study are available from the corresponding author on reasonable request.
Sequencing reads were mapped to the reference S288c genome (version R64, NCBI: GCF_000146045.2).

## Research involving human participants, their data, or biological material

Policy information about studies with human participants or human data. See also policy information about sex, gender (identity/presentation), and sexual orientation and race, ethnicity and racism.

| | |
|---|---|
| Reporting on sex and gender | Not applicable. |
| Reporting on race, ethnicity, or other socially relevant groupings | Not applicable. |
| Population characteristics | Not applicable. |
| Recruitment | Not applicable. |
| Ethics oversight | Not applicable. |

Note that full information on the approval of the study protocol must also be provided in the manuscript.

# Field-specific reporting

Please select the one below that is the best fit for your research. If you are not sure, read the appropriate sections before making your selection.

☒ Life sciences          ☐ Behavioural & social sciences          ☐ Ecological, evolutionary & environmental sciences

For a reference copy of the document with all sections, see nature.com/documents/nr-reporting-summary-flat.pdf

# Life sciences study design

All studies must disclose on these points even when the disclosure is negative.

| | |
|---|---|
| Sample size | No sample-size calculations were performed. Sample sizes were chosen to be comparable to published manuscripts within the field. Growth measurements and experimental evolution (doi: 10.1093/molbev/msaa172), live-cell microscopy (doi:10.1016/j.devcel.2019.01.018). |
| Data exclusions | For population growth rate measurements and single-cell microscopy experiments, experiments were only deemed successful if the control strains that were included in each experiment behaved as they should. 2 out of 14 population growth rate measurements were excluded (and later repeated successfully) because of this reason, and 0 microscopy experiments. For population growth rate measurements, single measurements for which the gcplyr package failed to calculate a growth rate (usually because of unexpected spikes in the growth curves due to e.g. bubble formation during incubation) were removed. For single cell microscopy, cells for which the SPB signal could not be reliably identified for 3 or more sequential frames were removed. From the resulting data, nothing was excluded from the analyses (e.g. no outliers were removed). All raw measurements can be found in the Source Data. |
| Replication | Population growth experiments were each set up with 16 independent replicates, 20 independent cells were chosen for each SPB tracking microscopy experiment, 50 cells were chosen for spindle curvature measurements, 56 independent replicate populations were established for each genotype in the evolution experiments, of which 1 clone from 4 (16chr), 4 (8chr), and 8 (3chr) populations were sequenced and 16 (16chr), 16 (8chr), and 56 (3chr) were checked for ploidy. Western blots were performed in duplicate using independent experiments (both successful). |
| Randomization | For population-level experiments, such as growth measurements, cell growth before Western blots, and experimental evolution, randomization is not relevant as yeast cultures are uniform on a population level. For single-cell microscopy experiments, random dividing cells were chosen for further analysis from overview images of asynchronous populations. |
| Blinding | Where possible, data were analyzed using semi-automated methods. In cases where manual filtering or correction was incorporated, same criteria were applied to all strains or treatments under investigation. During the experiments, blinding was not possible as each experiment was performed by an individual investigator who was aware of the experimental groups and treatments. |

# Reporting for specific materials, systems and methods

We require information from authors about some types of materials, experimental systems and methods used in many studies. Here, indicate whether each material, system or method listed is relevant to your study. If you are not sure if a list item applies to your research, read the appropriate section before selecting a response.

## Materials & experimental systems

| n/a | Involved in the study |
|---|---|
| ☐ | ☒ Antibodies |
| ☒ | ☐ Eukaryotic cell lines |
| ☒ | ☐ Palaeontology and archaeology |
| ☒ | ☐ Animals and other organisms |
| ☒ | ☐ Clinical data |
| ☒ | ☐ Dual use research of concern |
| ☒ | ☐ Plants |

## Methods

| n/a | Involved in the study |
|---|---|
| ☒ | ☐ ChIP-seq |
| ☒ | ☐ Flow cytometry |
| ☒ | ☐ MRI-based neuroimaging |

## Antibodies

| | |
|---|---|
| Antibodies used | Expansion microscopy: YL1/2 anti-α-tubulin (rat) (1:25),  gift from Gislene Pereira [COS Heidelberg, Germany]); goat anti-mouse-IgG coupled to Alexa Fluor 488 (Invitrogen A11029)(1:500). Western blot: mouse anti-HA antibody (Cat. No. 26183, Invitrogen)(1:1000), anti-actin antibody (Cat. No. MAB1501R, Chemicon)(1:1000), HRP-conjugated goat anti-mouse secondary antibody (Cat. No. 31430, Invitrogen)(1:10000). |
| Validation | Expansion microscopy: this antibody combination was successfully used for yeast in doi: 10.1242/jcs.260240.<br>Western blot: anti-HA: the antibody has been used successfully in Western blot and immunoprecipitation applications.(e.g. in budding yeast: doi: 10.26508/lsa.202201642); anti-actin: validated for use in ELISA, IC, IH, IH(P), WB for the detection of actin (e.g. in budding yeast: doi: 10.3390/ijms21249574); HRP goat anti-mouse: Product # 31430 has been successfully used in Western blot, IHC and IP applications (e.g. in budding yeast: doi: 10.3390/ijms21249574). |

## Plants

| | |
|---|---|
| Seed stocks | NA |
| Novel plant genotypes | NA |
| Authentication | NA |

