## [Peer Review File · Nature Cell Biology]

Peer Review Information

Journal: Nature Cell Biology

Manuscript Title: Spindle architecture constrains karyotype evolution

Corresponding author name(s): Dr Gautam Dey

Editorial Notes:

Reviewer Comments & Decisions:

Decision Letter, initial version:
--

Dear Gautam,

First, I sincerely apologize again for the delay in sharing our decision with you. I am very very sorry - we felt we needed to wait for Rev#3's comments given their expertise and we are truly sorry this delayed the process so much.

I also wished to send you and your colleagues my best wishes for the new year!

Thank you again for submitting your manuscript, "Spindle architecture constrains karyotype in budding yeast", to Nature Cell Biology. With our apologies for the time this has taken, your manuscript has now been seen by 3 referees, who are experts in cell division mechanisms (Referee #1); genome evolution (Referee #2); and synthetic genome biology (Referee #3). As you will see from their comments (attached below), they found the work of potential interest but have raised substantial concerns that in our view would need to be addressed with considerable revisions before we can consider publication in Nature Cell Biology.

As you may know, Nature Cell Biology editors discuss the referee reports in detail within the editorial team, including the chief editor, to identify key referee points that should be addressed with priority, and requests that are overruled as being beyond the scope of the current study. To guide the scope of the revisions, I have listed these points below. Our standard revision period is six months, and we are committed to providing a fair and constructive peer-review process, so please feel free to contact me if you would like to discuss any of the referee comments further or if you anticipate any issues or delays addressing the reviews.

In our view, it would be important to dedicate efforts in revision to address the comments below so as to strengthen the conclusions and interpretation:

A-- A priority in revision should be to add controls and analyses to convince these experts of the model, as Rev#1 in particular is worried that the data may also support the opposite interpretation, that the karyotype constrains spindle architecture:

Rev#1 points #1, #2, #3, #5

Rev#3 points #1, #2

B-- The reviewers asked for more description of the phenotypes that influence growth, and for more comprehensive descriptions:

Rev#1 point #4

Rev#3 point #3

C-- Two of the reviewers asked if you have done similar experiments in strains with one or with two chromosomes. While data addressing this point is not needed for publication, this is a question other readers may ask themselves, so we encourage you to discuss your rationale behind carrying out or not carrying out these experiments in the manuscript.

D-- Please also address the reviewers' minor points, requests for statistical analyses, discussion, and technical questions.

E-- Finally, please pay close attention to our guidelines on statistical and methodological reporting (listed below) as failure to do so may delay the reconsideration of the revised manuscript. In particular, please provide:

- a Supplementary Figure including unprocessed images of all gels/blots in the form of a multi-page pdf file. Please ensure that blots/gels are labeled and the sections presented in the figures are clearly indicated.
- a Supplementary Table including all numerical source data in Excel format, with data for different figures provided as different sheets within a single Excel file. The file should include source data giving rise to graphical representations and statistical descriptions in the paper and for all instances where the figures present representative experiments of multiple independent repeats, the source data of all repeats should be provided.

We would be happy to consider a revised manuscript that would satisfactorily address these points, unless a similar paper is published elsewhere, or is accepted for publication in Nature Cell Biology in the meantime.

- ensure that it conforms to our format instructions and publication policies (see below and <https://www.nature.com/nature/for-authors>).
- provide a point-by-point rebuttal to the full referee reports verbatim, as provided at the end of this letter.
- provide the completed Reporting Summary (found here <https://www.nature.com/documents/nr-reporting-summary.pdf>). This is essential for reconsideration of the manuscript will be available to editors and referees in the event of peer review. For more information see <http://www.nature.com/authors/policies/availability.html> or contact me.

When submitting the revised version of your manuscript, please pay close attention to our [href="https://www.nature.com/nature-portfolio/editorial-policies/image-integrity">Digital Image Integrity Guidelines](https://www.nature.com/nature-portfolio/editorial-policies/image-integrity). and to the following points below:

- that unprocessed scans are clearly labelled and match the gels and western blots presented in

figures.

-- that control panels for gels and western blots are appropriately described as loading on sample processing controls

-- all images in the paper are checked for duplication of panels and for splicing of gel lanes.

Nature Cell Biology is committed to improving transparency in authorship. As part of our efforts in this direction, we are now requesting that all authors identified as 'corresponding author' on published papers create and link their Open Researcher and Contributor Identifier (ORCID) with their account on the Manuscript Tracking System (MTS), prior to acceptance. ORCID helps the scientific community achieve unambiguous attribution of all scholarly contributions. You can create and link your ORCID from the home page of the MTS by clicking on 'Modify my Springer Nature account'. For more information please visit www.springernature.com/orcid.

This journal strongly supports public availability of data. Please place the data used in your paper into a public data repository, or alternatively, present the data as Supplementary Information. If data can only be shared on request, please explain why in your Data Availability Statement, and also in the correspondence with your editor. Please note that for some data types, deposition in a public repository is mandatory - more information on our data deposition policies and available repositories appears below.

[Redacted]

We hope that you will find our referees' comments and editorial guidance helpful. Please do not hesitate to contact me if there is anything you would like to discuss. Thank you again very much for considering NCB for your work,

Best wishes,

Melina

Melina Casadio, PhD
Senior Editor, Nature Cell Biology
ORCID ID: <https://orcid.org/0000-0003-2389-2243>

Reviewers' Comments:

Reviewer #1:

Remarks to the Author:

The manuscript by Helsen et al., addresses the interesting question of how the mitotic machinery adapts to karyotypic changes using chromosome engineering in budding yeast as a model system. Addressing this question is important, since karyotypic diversity is a hallmark of speciation and has also been implicated in human cancers. The main conclusion by the authors is that spindle architecture is the key rate-limiting factor that constrains ARTIFICIAL karyotype evolution in this system. I question whether the authors' own data is actually not suggesting the opposite, i.e. that karyotype constrains spindle architecture, as shown before in *Xenopus* (*laevis* vs. *tropicalis*; e.g. Loughlin et al., *Cell*. 147:1397-407, 2011). Indeed, the findings suggest that centromere/chromosome number/kinetochore-microtubule attachments are amongst the key rate-limiting factors themselves that might generate imbalance of spindle forces. Importantly, it remains vague what the critical rate-limiting factor is and experiments are suggested to clarify this point (see specific major points). Moreover, a more extensive discussion on the validity of these findings and potential limitations relative to other systems that evolved naturally would be a fair exercise to include. For instance, mother Nature gave us fission yeast with only 3 chromosomes (that remain mostly haploid in the wild), the main difference relative to budding yeast being the centromere/kinetochore size and respective number of attached microtubules. Yet, they segregate chromosomes just fine (likely due to 400 million years of evolutionary divergence between both species, as opposed to artificial evolution used in the present paper). Interestingly, when the authors added one extra mini-chromosome (centromere sequence only), this appeared to rescue fitness in an engineered low chromosome budding yeast strain generated by centromere excisions. As so, it seems that either chromosome or centromere/kinetochore number is directly or indirectly responsible to tolerate karyotype evolution, providing the required balance to sustain spindle forces, at least in budding yeast. Intriguingly, although it doesn't seem to be the case in budding yeast, in other species, including some mammals (e.g. Wang et al., *Science*, 377, 967-975, 2022), there is a clear limit on chromosome size for effective segregation. Thus, low chromosome number limit and how cells cope with this might be species-specific and evoke different mechanisms. This should be discussed and the study limitations acknowledged upfront. Overall, despite its limitations, this is a very elegant study that provides important cues on how dividing cells adapt to karyotype evolution. Specific points follow below.

Major issues:

1- One critical aspect related with yeast strains generated by chromosome engineering is that centromeres (and some telomeres) are excised during the fusion events. This implies that (at least) TWO variables are being manipulated in these experiments: chromosome number and centromere number. Importantly, by reducing centromere number in budding yeast, where each centromere forms a kinetochore that binds a single microtubule, this would cause quite a significant imbalance in the number of kinetochore microtubules relative to non-kinetochore microtubules and explains why alleviating microtubule pulling forces by non-kinetochore microtubules (either by low benomyl or KinI deletion) rescues normal fitness. More surprising is the fact that a single additional centromere/mini-chromosome also rescues fitness, suggesting that there is a critical threshold of centromere or chromosome number that ensures proper force balance in the spindle. To distinguish between these possibilities, the authors could envision to engineer a strain with low chromosome number (say with 3 chromosomes), while expanding centromere size on these chromosomes (e.g. two MT-binding units/fused chromosome). Alternatively, they could try to add 4 centromeric plasmids in the 1 chromosome cell and determine whether this also rescues fitness.

2- Related to this point, the authors use diploidization of haploid strains with 2 chromosomes to make the point about centromere/chromosome number and cell fitness. One important alteration often caused by alterations in ploidy is a proportional increase in spindle length that might contribute to accommodate/segregate more chromosomes, but the authors do not investigate whether this is the case in their situation. As so, it will be important to rule out that metaphase spindle length is not another factor that might mask the interpretation of the data in this experiment.

3- Another aspect that requires clarification is the timing of mitosis in the different strains. The authors use max spindle length in anaphase as reference, but since spindle architecture (bent and longer spindles) is disrupted in the low chromosome strains, this might not be a good reference. As so, why did the authors not consider anaphase onset as reference? Moreover, because fewer kinetochores to attach (and generate tension) would lead to faster SAC satisfaction in the low chromosome number strains, anaphase onset and spindle elongation might start prematurely. Clarification of this point will be critical to interpret the Mad2deletion experiment because if anaphase is indeed starting prematurely, SAC inactivation would not make a big difference in mitotic timing in the low chromosome number strains. Some direct readout of mitotic timing should be included (Cyclin B?). Moreover, it seems that the differences in mitotic timing between 16 or 3 chromosome strains (fig. 1f) are heavily influenced by that ONE cell that started SPB separation much earlier. The authors should consider analysing median rather than mean distributions.

4- The authors refer to other defects, such as atypical nuclear distortions and spindle displacement, but do not provide a quantitative account for these defects. An estimate of frequency relative to other strains would be informative.

5- A critical related question that is not at all addressed is about chromosome segregation efficiency in the low chromosome number strain. Lower fitting in this strain might reflect non-mitotic causes (e.g. DNA replication, DNA damage, telomere length and number, etc) derived from the chromosome engineering. The authors have looked at Ndc80-labelled cells to illustrate the apparently higher dispersion of chromosomes in the low chromosome number strain, but do not comment on whether chromosomes actually segregate accurately. An account of chromosome segregation fidelity must be provided. Information from their single cell sequencing analysis might also shed light into this aspect.

Minor issues:

1- Fig3b: haploid strain with 2 chromosomes seems not to have a growth disadvantage relative to the 16 chromosome strain, as opposed to the strain with 3 chromosomes. Please comment.

2- Could the authors comment on the formation and origin of triploid cells exclusively in the intermediate chromosome number strain with 8 chromosomes?

3- Regarding the mutations in the evolved strains, could the authors comment on the functions of the mutated genes? The genes might be different, but participate in the same process.

Reviewer #2:

Remarks to the Author:

This paper by Helsen et al provides a series of interesting insights into the behavior of engineered karyotype constructed previously by stringing together chromosomes end to end. Strains varying in haploid number were studied. The paper describes data leading to the following conclusions: 1) the spindle pole bodies of these megachromosomes are smaller than those of strains with the conventional 16 chromosomes 2) the spindles show a great deal of flexibility by live imaging, apparently lengthening and bending to accommodate the much larger chromosome arms 3) Diploidy arises during experimental evolution of these strains, and is shown to suppress the defects associated

with long spindles presumably as a consequence of providing more room to accommodate distorted spindles. 4) One very nice part of this study is that something important and rather specific happens to the spindle at the boundary between four and five centromeres, as shown by adding single CEN plasmid to a strain with $n=4$. Mechanistically, the results point to defects in outward forces in the spindle in strains with less than 5 centromeres, which triggers the spindle assembly checkpoint. Remarkably, these strains benefit from the combination of a *mad2* deletion in the presence of small amounts of benomyl. The results point to the chromosome segregation machinery being able to accommodate a wide variety of karyotypes, despite the fact that native *S cerevisiae* isolates do not show anything like the karyotype variations studied here, which is a bit surprising.

I wonder if the authors ever deployed their assays on the 1 megachromosome strain from the Qin lab and what the results were.

Fig 5c. There are no annotations of statistical significance in this chart. Is that because the results are not of statistical significance or is this an omission?

Reviewer #3:

Remarks to the Author:

In the manuscript by Jana Helsen et al, titled Spindle architecture constrains karyotype in budding yeast, the authors using cell biological profiling, genetic engineering, and experimental evolution to uncover the underlining mechanism for observed mitotic growth defect in strains with fewer number of chromosomes. They established an inherent link between karyotype and the cell division machinery during evolution, and provides insight into how the mechanics of a core cellular process can determine the limitations of evolution. The experiments were designed logically and executed nicely, and the manuscript is well-written. The overall conclusion is supported by the experimental results. and I really enjoyed reading the manuscript.

I have three major concerns:

1. The growth rate of wt (16 chr) is around 0.46 and that of 4/3 chr is around 0.42 in haploid cells. These numbers seem not stable and changed in different experiments if comparing Figure 1b with Figure 3a/3b. It makes the statistically analysis less convincing. Particularly, although the authors firmly stated that addition of CEN plasmid rescue the growth defect, Fig3a only showed very subtle changes. In addition, besides only use the CEN plasmid, it should also include more controls such as a 2-micron plasmid (without additional centromere) or two CEN plasmids with different markers (more CEN in the 3 chr background). Maybe the latter could make the result clearer. Are there other ways to alter the number of spindle microtubules? If there are, it will provide additional convincing evidences and make the conclusion more reliable.
2. The claim on "the force imbalance causes kinetochore declustering and triggers the SAC", seems not fully supported. Statistically analysis of the declustering is required. On the other hand, besides using the *Mad2* mutant, additional evidence of activation of SAC is needed.
3. Besides the spindle segregation phenotype, are there any other potential causes for the growth defects such as activation of cell cycle checkpoint etc. An survey on these potentials at the begin will enhance the quality of this manuscript

Minor concerns:

1. Did you try the strain with only 1 or 2 chromosomes? I am curious how these strains behave in the

- analysis. It is still a mystery why the 2 chromosomes can't be fused into 1 in Luo's paper. (This is not required for the publication of this manuscript)
2. To rule out potential effect from double the gene content, overall transcription analysis might be useful in both haploid and diploid strains in Fig2
 3. Fig4b/4g, color is too close. It's hard to distinguish different samples
 4. Adding analysis of epistatic effects between #of chromosome and the SAC mutants, similar to Fig4f
 5. Fig5c, adding p-value or *

Methods should be written concisely, but should contain all elements necessary to allow interpretation and replication of the results. As a guideline, Methods sections typically do not exceed 3,000 words. The Methods should be divided into subsections listing reagents and techniques. When citing previous methods, accurate references should be provided and any alterations should be noted. Information must be provided about: antibody dilutions, company names, catalogue numbers and clone numbers for monoclonal antibodies; sequences of RNAi and cDNA probes/primers or company names and catalogue numbers if reagents are commercial; cell line names, sources and information on cell line identity and authentication. Animal studies and experiments involving human subjects must be reported in detail, identifying the committees approving the protocols. For studies involving human subjects/samples, a statement must be included confirming that informed consent was obtained. Statistical analyses and information on the reproducibility of experimental results should be provided in a section titled "Statistics and Reproducibility".

All Nature Cell Biology manuscripts submitted on or after March 21 2016 must include a Data availability statement as a separate section after Methods but before references, under the heading "Data Availability". For Springer Nature policies on data availability see <http://www.nature.com/authors/policies/availability.html>; for more information on this particular policy see <http://www.nature.com/authors/policies/data/data-availability-statements-data-citations.pdf>. The Data availability statement should include:

- Accession codes for primary datasets (generated during the study under consideration and

designated as "primary accessions") and secondary datasets (published datasets reanalysed during the study under consideration, designated as "referenced accessions"). For primary accessions data should be made public to coincide with publication of the manuscript. A list of data types for which submission to community-endorsed public repositories is mandated (including sequence, structure, microarray, deep sequencing data) can be found here <http://www.nature.com/authors/policies/availability.html#data>.

- Unique identifiers (accession codes, DOIs or other unique persistent identifier) and hyperlinks for datasets deposited in an approved repository, but for which data deposition is not mandated (see here for details <http://www.nature.com/sdata/data-policies/repositories>).
- At a minimum, please include a statement confirming that all relevant data are available from the authors, and/or are included with the manuscript (e.g. as source data or supplementary information), listing which data are included (e.g. by figure panels and data types) and mentioning any restrictions on availability.
- If a dataset has a Digital Object Identifier (DOI) as its unique identifier, we strongly encourage including this in the Reference list and citing the dataset in the Methods.

We recommend that you upload the step-by-step protocols used in this manuscript to the Protocol Exchange. More details can found at www.nature.com/protocolexchange/about.

All imaging data should be accompanied by scale bars, which should be defined in the legend. Cropped images of gels/blots are acceptable, but need to be accompanied by size markers, and to retain visible background signal within the linear range (i.e. should not be saturated). The boundaries of panels with low background have to be demarked with black lines. Splicing of panels should only be considered if unavoidable, and must be clearly marked on the figure, and noted in the legend with a statement on whether the samples were obtained and processed simultaneously. Quantitative comparisons between samples on different gels/blots are discouraged; if this is unavoidable, it should only be performed for samples derived from the same experiment with gels/blots were processed in parallel, which needs to be stated in the legend.

Figures should be provided at approximately the size that they are to be printed at (single column is 86 mm, double column is 170 mm) and should not exceed an A4 page (8.5 x 11"). Reduction to the scale that will be used on the page is not necessary, but multi-panel figures should be sized so that the whole figure can be reduced by the same amount at the smallest size at which essential details in each panel are visible. In the interest of our colour-blind readers we ask that you avoid using red and green for contrast in figures. Replacing red with magenta and green with turquoise are two possible

colour-safe alternatives. Lines with widths of less than 1 point should be avoided. Sans serif typefaces, such as Helvetica (preferred) or Arial should be used. All text that forms part of a figure should be rewritable and removable.

SUPPLEMENTARY INFORMATION – Supplementary information is material directly relevant to the conclusion of a paper, but which cannot be included in the printed version in order to keep the manuscript concise and accessible to the general reader. Supplementary information is an integral

part of a Nature Cell Biology publication and should be prepared and presented with as much care as the main display item, but it must not include non-essential data or text, which may be removed at the editor's discretion. All supplementary material is fully peer-reviewed and published online as part of the HTML version of the manuscript. Supplementary Figures and Supplementary Notes are appended at the end of the main PDF of the published manuscript.

The total number of Supplementary Figures (not including the "unprocessed scans" Supplementary Figure) should not exceed the number of main display items (figures and/or tables (see our Guide to Authors and March 2012 editorial <http://www.nature.com/ncb/authors/submit/index.html#suppinfo>; <http://www.nature.com/ncb/journal/v14/n3/index.html#ed>). No restrictions apply to Supplementary Tables or Videos, but we advise authors to be selective in including supplemental data.

GUIDELINES FOR EXPERIMENTAL AND STATISTICAL REPORTING

REPORTING REQUIREMENTS – We are trying to improve the quality of methods and statistics reporting in our papers. To that end, we are now asking authors to complete a reporting summary that collects information on experimental design and reagents. The Reporting Summary can be found here <https://www.nature.com/documents/nr-reporting-summary.pdf> If you would like to reference the guidance text as you complete the template, please access these flattened versions at <http://www.nature.com/authors/policies/availability.html>.

STATISTICS – Wherever statistics have been derived the legend needs to provide the n number (i.e. the sample size used to derive statistics) as a precise value (not a range), and define what this value represents. Error bars need to be defined in the legends (e.g. SD, SEM) together with a measure of centre (e.g. mean, median). Box plots need to be defined in terms of minima, maxima, centre, and percentiles. Ranges are more appropriate than standard errors for small data sets. Wherever statistical significance has been derived, precise p values need to be provided and the statistical test used needs to be stated in the legend. Statistics such as error bars must not be derived from $n < 3$. For

sample sizes of $n < 5$ please plot the individual data points rather than providing bar graphs. Deriving statistics from technical replicate samples, rather than biological replicates is strongly discouraged. Wherever statistical significance has been derived, precise p values need to be provided and the statistical test stated in the legend.

Author Rebuttal to Initial comments

Reviewer #1:

Remarks to the Author:

The manuscript by Helsen et al., addresses the interesting question of how the mitotic machinery adapts to karyotypic changes using chromosome engineering in budding yeast as a model system. Addressing this question is important, since karyotypic diversity is a hallmark of speciation and has also been implicated in human cancers. The main conclusion by the authors is that spindle architecture is the key rate-limiting factor that constrains ARTIFICIAL karyotype evolution in this system. I question whether the authors' own data is actually not suggesting the opposite, i.e. that karyotype constrains spindle architecture, as shown before in *Xenopus* (*laevis* vs. *tropicalis*; e.g. Loughlin et al., Cell. 147:1397-407, 2011).

*In our experiments, spindle architecture remains constant (i.e., one kinetochore microtubule per centromere, and one centromere per chromosome), and our cells evolve to change their karyotype through diploidization. Cells with fewer than 5 centromeres have a fitness defect and evolutionary disadvantage and as a result, evolution towards such karyotypes will be constrained as a direct result of how the budding yeast spindle is set up. Thus, on the evolutionary timescale used in our experiments, we are quite certain about the directionality of our conclusion. On larger evolutionary timescales, both could be true. We do think it important to note that karyotypes are known to evolve much faster than spindle proteins. For example, many other budding yeasts (comprising different genera) have the same spindle architecture as *S. cerevisiae*, but their karyotypes vary significantly (between $n=6$ and $n=16$). Our results could help explain why $n=6$ seems to be the lower limit for these species.*

Indeed, the findings suggest that centromere/chromosome number/kinetochore-microtubule attachments are amongst the key rate-limiting factors themselves that might generate imbalance of spindle forces. Importantly, it remains vague what the critical rate-limiting factor is and experiments are suggested to clarify this point (see specific major points). Moreover, a more extensive discussion on the validity of these findings and potential limitations relative to other systems that evolved naturally would be a fair exercise to include. For instance, mother Nature gave us fission yeast with only 3 chromosomes (that remain mostly haploid in the wild), the main difference relative to budding yeast being the centromere/kinetochore size and respective number of attached microtubules. Yet, they segregate chromosomes just fine (likely due to 400 million years of evolutionary divergence between both species, as opposed to artificial evolution used in the present paper). Interestingly, when the authors added one extra mini-chromosome (centromere sequence only), this appeared to rescue fitness in an engineered low chromosome budding yeast strain generated by centromere excisions. As so, it seems that either chromosome or centromere/kinetochore number is directly or indirectly responsible to tolerate karyotype evolution, providing the required balance to sustain spindle forces, at least in budding yeast. Intriguingly, although it doesn't seem to be the case in budding yeast, in other species, including some mammals (e.g. Wang et al., *Science*, 377, 967-975, 2022), there is a clear limit on chromosome size for effective segregation. Thus, low chromosome number limit and how cells cope with this might be species-specific and evoke different mechanisms. This should be discussed and the study limitations acknowledged upfront. Overall, despite its limitations, this is

a very elegant study that provides important cues on how dividing cells adapt to karyotype evolution. Specific points follow below.

We thank the reviewer for their very helpful comments, and believe that by addressing them our manuscript has become more clear and nuanced. Specifically, we:

- Make it more clear, and provide additional proof that the number of centromeres/kMT attachments is the rate-limiting factor, rather than the number of chromosomes (see also our response to comment 1) (paragraph on p6, specifically lines 2-8, 20-22, 33-35).*
- Provide a more extensive discussion on how our findings fit into a wider evolutionary context, by comparing spindle (and centromere) architecture in budding yeast (1 kMT / centromere / chromosome) with other species (e.g. fission yeast with regional centromeres and more attachments) (p11, lines 31-34 and 40-43). We also briefly discuss the effect of chromosome size on efficient segregation and why this is probably not a limiting factor in budding yeast (p6 lines 24-29).*
- Through additional experiments, we strengthen our conclusion that the delay is metaphase-specific and triggers the SAC. We also quantified the kinetochore declustering phenotype.*
- Address the reviewer's other concerns below in detail.*

Major issues:

1- One critical aspect related with yeast strains generated by chromosome engineering is that centromeres (and some telomeres) are excised during the fusion events. This implies that (at least) TWO variables are being manipulated in these experiments: chromosome number and centromere number. Importantly, by reducing centromere number in budding yeast, where each centromere forms a kinetochore that binds a single microtubule, this would cause quite a significant imbalance in the number of kinetochore microtubules relative to non-kinetochore microtubules and explains why alleviating microtubule pulling forces by non-kinetochore microtubules (either by low benomyl or KinI deletion) rescues normal fitness. More surprising is the fact that a single additional centromere/mini-chromosome also rescues fitness, suggesting that there is a critical threshold of centromere or chromosome number that ensures proper force balance in the spindle. To distinguish between these possibilities, the authors could envision to engineer a strain with low chromosome number (say with 3 chromosomes), while expanding centromere size on these chromosomes (e.g. two MT-binding units/fused chromosome). Alternatively, they could try to add 4 centromeric plasmids in the 1 chromosome cell and determine whether this also rescues fitness.

To distinguish between the effect of chromosome number and centromere number, we now included an orthogonal approach to provide additional evidence that it is in fact the additional CEN that rescues the strain with four chromosomes, and not chromosome number per se. We transformed a small (9.7 kb) artificial chromosome into the same strains, which also contains one centromere, but in contrast to the centromeric plasmid is linear and has telomeres. We observe the same exact trend as for the centromeric plasmid: adding the artificial chromosome rescues

the growth defect in the strain with four chromosomes, but not in the strain with three chromosomes:

New Figure 3a | Maximum growth rates of fused-chromosome strains with and without additional centromeric plasmid or artificial chromosome. Boxes represent means and standard deviation. Means were compared using a Student's t-test. Red circles indicate centromeres.

The new data were also added to the summary of epistatic effects (Figure 4f):

New Figure 4f | Summary of epistatic effects of different perturbations. Both diploidization, adding a centromeric plasmid, and an artificial chromosome increase the inward force, and adding benomyl or deleting KIP1 decrease the outward force. Boxes represent means and standard deviation.

Together with our experiments in diploids (we now also have single cell microscopy for these strains, see response to comment 2), these observations indicate that it is not the number of chromosomes (plasmid and artificial chromosome have the same effect), the amount of DNA (diploids have double the amount of DNA and the cut-off is still 4 CENs total), but really the total number of CENs in the cell that determines whether the cell has a growth/mitotic defect.

Another factor to consider is that adding the extra CEN also rescues the mitotic phenotypes. In the updated version of the manuscript, we included new versions of Fig. 4b and Fig. 4g to more clearly highlight and explain one of these phenotypes: the steady increase in SPB distance during metaphase in the fusion strains. Fig. 4b and 4g now show the distance between SPBs at the end

of metaphase instead to make it easier to compare between the different strains and treatments. One thing we would like to point out in response to the reviewer's concern here, is that adding the centromeric plasmid in a four-chromosome strain reduces this distance to the level we see in a 5-chromosome strain. This is another indication that the number of centromeres, through kMT attachments, is the limiting factor rather than the number of chromosomes.

New Figure 4b and 4g | (b) Distance between SPBs at the end of metaphase for different fusion strains. Boxes represent means and standard deviation. (g) Distance between SPBs at the end of metaphase for different fusion strains and the effects of increasing inward force (+ pCEN) or decreasing outward force (+ benomyl). Means were compared using a Student's t-test; * $p < 0.05$, *** $p < 0.001$.

In the manuscript, we did originally also argue that the number of centromeres (through the number of kMT attachments) is in fact the rate-limiting factor, rather than the number of chromosomes, but we now make this more explicit in the text (paragraph on p6, specifically lines 2-8, 20-22, 33-35). In the discussion, we also added a section on how centromere number relates to chromosome number in other species (p11, lines 31-34 and 40-43).

Note: in budding yeast, cells with dicentric chromosomes are known to be less fit since these cells have no way of ensuring that kMTs are attached to the same SPB. As a result, 50% of the time the two centromeres will be attached to opposite spindle pole bodies, leading to chromosome breakage (e.g. Brock and Bloom J. Cell Sci. 1994).

2- Related to this point, the authors use diploidization of haploid strains with 2 chromosomes to make the point about centromere/chromosome number and cell fitness. One important alteration often caused by alterations in ploidy is a proportional increase in spindle length that might contribute to accommodate/segregate more chromosomes, but the authors do not investigate whether this is the case in their situation. As so, it will be important to rule out that metaphase spindle length is not another factor that might mask the interpretation of the data in this experiment.

We constructed new diploid strains with tagged SPBs and quantified the mitotic defect in strains with $n = 16, 8, 3$, and 2 chromosomes to show that both the growth rescue we observed for the

diploid 3 chromosome strain and the growth defect of the diploid 2 chromosome strain (Fig. 3b) are indeed a mitotic phenotype (new Fig. 3e and new Extended Data Figure 3).

New Figure 3e | The time from SPB doubling to max. anaphase separation for diploid strains. Boxes represent the means and standard deviation. Means were compared using a Student's *t*-test; * $p < 0.05$.

New Extended Data Figure 3 | Distance between SPBs over time for diploidized strains. For normalisation, the time point with maximal SPB separation during anaphase was set to zero.

Even though spindle and cell size are indeed different (new Extended Data Figure 3), here too, we observe both a growth defect and mitotic defect from #CEN = 4 and we observe the same steady increase in SPB distance during metaphase in the fusion strains (new Extended Data Figure 4a).

a

New Extended Data Figure 4a | Distance between SPBs at the end of metaphase for different diploid fusion strains. Boxes represent means and standard deviation.

3- Another aspect that requires clarification is the timing of mitosis in the different strains. The authors use max spindle length in anaphase as reference, but since spindle architecture (bent and longer spindles) is disrupted in the low chromosome strains, this might not be a good reference. As so, why did the authors not consider anaphase onset as reference? Moreover, because fewer kinetochores to attach (and generate tension) would lead to faster SAC satisfaction in the low chromosome number strains, anaphase onset and spindle elongation might start prematurely. Clarification of this point will be critical to interpret the Mad2deletion experiment because if anaphase is indeed starting prematurely, SAC inactivation would not make a big difference in mitotic timing in the low chromosome number strains. Some direct readout of mitotic timing should be included (Cyclin B?). Moreover, it seems that the differences in mitotic timing between 16 or 3 chromosome strains (fig. 1f) are heavily influenced by that ONE cell that started SPB separation much earlier. The authors should consider analysing median rather than mean distributions.

Instead of spindle length, we use the maximum separation between SPBs as a reference. That way, bent spindles won't affect our measurements. Using spindle lengths would make our measurements more noisy, especially in anaphase during which bends are more prevalent. However, both methods should give us the same values with regards to estimating the time of max. anaphase separation. Importantly, another reason why we use SPBs instead of spindles is to give us the time of SPB doubling. We do not use anaphase onset as a reference, because finding that exact time point is very hard in the fused strains since the spindle already elongates during metaphase. On top of the natural noise in the measurements, the transition to anaphase is sometimes hard to pinpoint on a cell-to-cell basis. By contrast, the time of max. anaphase separation is much less ambiguous.

To have a more direct readout of mitotic timing and SAC activation, we followed Pds1 (securin) levels upon release from a G1 arrest. During metaphase, securin binds and inhibits separase, the protease that degrades cohesin. During the metaphase-to-anaphase transition, securin is degraded, releasing separase so cohesin can be degraded and sister chromatids unlinked. SAC activation stabilises Pds1, and therefore prevents this transition. We switched the mating type of our strains (to allow for G1 arrest with alpha factor), and tagged Pds1 with a 3xHA tag. After

release from G1 arrest, we collected samples at specific timepoints, and performed a Western blot to quantify Pds1 levels (new Figure 5d and new Extended Data Figure 5f).

New Figure 5d | Western blot analysis of Pds1 levels after G1 release for 16- and 3-chromosome strains. Cells were collected at the indicated time points and alpha factor was added again 45 minutes after release to prevent the cells from entering a second cell cycle. Actin levels were used as a loading control. Pds1-normalised values are shown in the bar plot at the bottom.

New Extended Data Figure 5f | Western blot analysis of Pds1 levels after G1 release for 16- and 3-chromosome strains, focussed on the second cell cycle after release. Cells were collected at the indicated time points. Ponceau S staining was used as a loading control. Pds1-normalised values are shown in the bar plot at the bottom.

Consistent with both our population growth experiments and single cell microscopy - both showing a 5-10 minute delay in growth rate and duration of mitosis respectively - the 3-chromosome strain displays higher Pds1 levels at the end of the cell cycle compared to the wild type, indicating that metaphase and SAC activation persist longer in a significant proportion of the population. As expected for a 5-10 minute delay, the difference is small (~15% higher Pds1 levels in 3-chromosome strain at the relevant time points), but consistent across independent experiments and can be observed in both the first and second cell cycle after G1 release.

Even though we have sufficient evidence to show that the growth defect is the result of a mitotic defect, caused by excess outward force in the spindle which in turn triggers the SAC, we do not know the exact molecular underpinnings of this last step. Possibly, since the metaphase spindle elongates too rapidly, due to a lack of inward forces, chromosomes fail to biorient efficiently and cannot generate tension across kinetochores. This in turn would lead to a metaphase delay through activation of the SAC. One piece of evidence that is consistent with this hypothesis is that kinetochores are declustered in the 3-chromosome strain during metaphase. When the outward force in the metaphase is decreased by adding benomyl, the kinetochore signal becomes more bilobed (i.e. bimodal Ndc80 signal along the spindle pole to spindle pole axis). Our manuscript now contains a quantitative analysis of the declustering phenotype (new Figure 5e).

New Figure 5e | Proportion of cells with a bilobed kinetochore signal (i.e. bimodal Ndc80 signal along the spindle pole to spindle pole axis) in both metaphase and anaphase. Proportions were compared using a two-proportions Z-test; * $p < 0.05$.

We adjusted our discussion on this hypothesis in the updated manuscript (p11 lines 15-25).

Finally, for Fig1f, we only showed data from one experiment, but we now included data from other, independent experiments to better resolve the variation in mitotic timing for both the 4- and 3-chromosome strain. We find another 'outlier', and opted to keep the outliers in our figure. The 2 outliers have very little influence on the significance of the test ($p = 0.0002993$ with outliers, $p = 0.0006188$ without outliers).

*Updated Figure 1f | The time from SPB doubling to max. anaphase separation. Boxes represent the mean and standard deviation. Means were compared using a Student's t-test; * $p < 0.05$, *** $p < 0.001$.*

4- The authors refer to other defects, such as atypical nuclear distortions and spindle displacement, but do not provide a quantitative account for these defects. An estimate of frequency relative to other strains would be informative.

To more effectively focus on the growth and mitotic defects, which, with the additional data in the updated manuscript have become even more central to our story, we have moved the spindle defects (bending and displacement) shown in the previous version of figure 1 to the supplementary materials. Specifically:

- The 'bent' spindle phenotype is observed during anaphase, and we now conclusively prove that the growth defect/mitotic delay is the result of a delay in metaphase.*
- The spindle displacement phenotype is rare (we observed it in 2 out of the 30 cells in which we followed the tub1 fluorescence over time for a full mitotic cycle), making it hard to accurately quantify without following a very large number of cells. Additionally, we now show that the spindle positioning checkpoint (SPOC) - the checkpoint that responds to such displacements of the spindle - is not involved in the growth defect (see new figure in reply to next comment).*
- The atypical distortions of the nucleus are present in all (100%) 3-chromosome cells undergoing mitosis, and are still present in the diploids. As such, while interesting, we also do not believe that this phenotype is especially relevant for our conclusions.*

Instead, we now also have a new Figure 1c with a montage of SPBs over time to accompany and clarify the next two panels.

5- A critical related question that is not at all addressed is about chromosome segregation efficiency in the low chromosome number strain. Lower fitting in this strain might reflect non-mitotic causes (e.g. DNA replication, DNA damage, telomere length and number, etc) derived from the chromosome engineering. The authors have looked at Ndc80-labelled cells to illustrate

the apparently higher dispersion of chromosomes in the low chromosome number strain, but do not comment on whether chromosomes actually segregate accurately. An account of chromosome segregation fidelity must be provided. Information from their single cell sequencing analysis might also shed light into this aspect.

We do not think the fitness defect reflects a non-mitotic cause or a decrease in chromosome segregation fidelity, for the following reasons:

- *The fitness defect and mitotic delay are both **completely** fixed both by all our perturbations and by evolving our cells in the lab, and we showed that the defect reflects a delay in metaphase. It is important to keep in mind that evolution experiments are a powerful way of finding the largest contributor to fitness defects from an evolutionary point of view. As stated in the manuscript, we find diploidization to be the only adaptive mechanism, and find no mutations linked to chromosome segregation fidelity (see also list of mutations with updated gene function descriptions).*
- *The transcriptomic analysis performed by the authors who originally constructed the strains shows not a single differentially expressed gene involved in chromosome segregation fidelity.*
- *Using the artificial chromosome described above, we now have a way of controlling for telomere length/number. Adding a centromeric piece of DNA without telomeres (plasmid) has the same effect as adding one with telomeres (the YAC).*
- *To verify that other checkpoints apart from the SAC are not involved, we now also test whether deleting a component of the DNA damage checkpoint ($RAD9\Delta$), increasing DNA damage (+hydroxyurea), or deleting a component of the spindle positioning checkpoint ($BUB2\Delta$) influence the defect in the 3-chromosome strain (new Extended Data Figure 5b-d). None of these perturbations alleviate (or exacerbate) the defect.*

*New Extended Data Figure 5b-d | (b) Maximum growth rates of fused-chromosome strains with and without $RAD9$ deletion. Boxes represent means and standard deviations. Means were compared using a Student's t -test; ** $p < 0.01$. (c) Maximum growth rates of fused-chromosome strains with and without 10 mM hydroxyurea. Boxes represent means and standard deviations. Means were compared using a Student's t -test; *** $p < 0.001$. (d) Maximum growth rates of fused-chromosome strains with and without $BUB2$ deletion. Boxes represent means and standard deviations. Means were compared using a Student's t -test; *** $p < 0.001$.*

- The Western blots shown above show that there is no delay in S phase in 3-chromosome cells.
- We expect chromosome missegregation to have a much more pronounced effect on fitness in 3-chromosome cells compared to wild-type cells, since it would lead to an extra copy of $\sim\frac{1}{3}$ of the genome instead of $\sim\frac{1}{16}$ th (and the other daughter cell would be missing $\sim\frac{1}{3}$ of its genome). This would lead to delays throughout the cell cycle, not just metaphase, or an increase in the amount of dead cells, neither of which we observe.
- Although we observe a higher dispersion of kinetochores during metaphase in the 3-chromosome strain, this gets resolved during anaphase (see original montages in Figure 5 and new quantification in Figure 5e).

Minor issues:

1- Fig3b: haploid strain with 2 chromosomes seems not to have a growth disadvantage relative to the 16 chromosome strain, as opposed to the strain with 3 chromosomes. Please comment.

In Fig 3b, the growth defect of the haploid strain with 2 chromosomes is the same as the defect seen in the haploid strain with 3 chromosomes (average growth rate is nearly identical), but the 2-chromosome strain shows more variation, decreasing the statistical significance of the difference. However, in independent growth experiments, we observe the same trend. For the experiment shown in Extended Data Figure 1a, we actually have additional data we originally chose not to show for simplicity's sake. In the new version of this figure, we reintroduced the data for 5, 4, and 2 chromosomes. In this experiment, the average growth rate of the 2-chromosome strain is again similar to that of the 4- and 3-chromosome strain, but this time the difference is significant. We could introduce these additional data in Fig 3b, but in general we prefer to avoid combining measurements from different experiments in the same plots, due to slight differences in absolute average growth rates between experiments.

New Extended Data Figure 1a | Maximum growth rates of fused-chromosome strains on synthetic complete medium with 2% dextrose (SCD). Boxes show the means and standard deviation. Means were compared using a Student's t-test; * $p < 0.05$.

2- Could the authors comment on the formation and origin of triploid cells exclusively in the intermediate chromosome number strain with 8 chromosomes?

We think triploidization, like diploidization, can sometimes occur by chance. Anecdotally, we have observed a low frequency of triploids and even tetraploids in other evolution experiments, none of which could be linked back to the ancestral genotype. Often, researchers chose to only follow up and report on evolved haploids since all their mutations are 'homozygous', making it much easier to identify beneficial mutations.

3- Regarding the mutations in the evolved strains, could the authors comment on the functions of the mutated genes? The genes might be different, but participate in the same process.

To account for recurrent mutations in the same process, we used our lists of mutations as input in different GO enrichment tools (PANTHER, GOzilla). However, not a single GO term was enriched (no 'biological process', no 'molecular function', nor 'cellular component'). We now include an additional column in Extended Data Table 5 (list of mutations), with a short description of the mutated gene's function.

Reviewer #2:

Remarks to the Author:

This paper by Helsen et al provides a series of interesting insights into the behavior of engineered karyotype constructed previously by stringing together chromosomes end to end. Strains varying in haploid number were studied. The paper describes data leading to the following conclusions: 1) the spindle pole bodies of these megachromosomes are smaller than those of strains with the conventional 16 chromosomes 2) the spindles show a great deal of flexibility by live imaging, apparently lengthening and bending to accommodate the much larger chromosome arms 3) Diploidy arises during experimental evolution of these strains, and is shown to suppress the defects associated with long spindles presumably as a consequence of providing more room to accommodate distorted spindles. 4) One very nice part of this study is that something important and rather specific happens to the spindle at the boundary between four and five centromeres, as shown by adding single CEN plasmid to a strain with $n=4$. Mechanistically, the results point to defects in outward forces in the spindle in strains with less than 5 centromeres, which triggers the spindle assembly checkpoint. Remarkably, these strains benefit from the combination of a *mad2* deletion in the presence of small amounts of benomyl. The results point to the chromosome segregation machinery being able to accommodate a wide variety of karyotypes, despite the fact that native *S cerevisiae* isolates do not show anything like the karyotype variations studied here, which is a bit surprising.

I wonder if the authors ever deployed their assays on the 1 megachromosome strain from the Qin lab and what the results were.

We first planned to also use the single-chromosome strain in our analyses, but when handling the strain it quickly became apparent that it is more unstable than the other strains and not suited for our type of analyses: unevolved, log-phase populations that are started from our frozen stock already show extensive variability in cell size:

Scale = 10 μ m

We therefore decided to focus on an 'intermediate' strain (the 3-chromosome strain), since it has a clear growth defect and mitotic defect, while being much more stable and showing less intrapopulation variability.

Fig 5c. There are no annotations of statistical significance in this chart. Is that because the results are not of statistical significance or is this an omission?

Statistical values have been added.

Reviewer #3:

Remarks to the Author:

In the manuscript by Jana Helsen et al, titled Spindle architecture constrains karyotype in budding yeast, the authors using cell biological profiling, genetic engineering, and experimental evolution to uncover the underlining mechanism for observed mitotic growth defect in strains with fewer number of chromosomes. They established an inherent link between karyotype and the cell division machinery during evolution, and provides insight into how the mechanics of a core cellular process can determine the limitations of evolution. The experiments were designed logically and executed nicely, and the manuscript is well-written. The overall conclusion is supported by the experimental results. and I really enjoyed reading the manuscript.

I have three major concerns:

1. The growth rate of wt (16 chr) is around 0.46 and that of 4/3 chr is around 0.42 in haploid cells. These numbers seem not stable and changed in different experiments if comparing Figure 1b with Figure 3a/3b. It makes the statistically analysis less convincing. Particularly, although the authors firmly stated that addition of CEN plasmid rescue the growth defect, Fig3a only showed very subtle changes. In addition, besides only use the CEN plasmid, it should also include more controls such as a 2-micron plasmid (without additional centromere) or two CEN plasmids with different markers (more CEN in the 3 chr background). Maybe the latter could make the result clearer. Are there other ways to alter the number of spindle microtubules? If there are, it will provide additional convincing evidences and make the conclusion more reliable.

Absolute growth rates can differ slightly between experiments. To try and minimise these fluctuations, we used the same batch of medium (frozen down) for all the evolution, microscopy and growth rate experiments. In addition, all cultures for growth rate experiments and microscopy were started from a log-phase preculture to minimise the effect of the growth phase on the ensuing growth dynamics. Since we know that even then, growth rates sometimes still fluctuate slightly between experiments (probably because of small fluctuations in temperature, shaking, oxygenation, or a combination thereof), we always included two control strains in each experiment (ancestral wild type and ancestral 3-chromosome strain), to verify that the relative measurements and the relative growth defect were stable and consistent across experiments.

For Figure 3a, we now included an orthogonal approach to provide more evidence that an additional CEN can rescue the strain with four chromosomes. We transformed a small (9.7 kb) artificial chromosome into the same strains, which also contains one centromere, but in contrast to the centromeric plasmid is linear and has telomeres. We observe the same exact trend as for the centromeric plasmid: adding the artificial chromosome rescues the growth defect in the strain with four chromosomes, but not in the strain with three chromosomes:

New Figure 3a | Maximum growth rates of fused-chromosome strains with and without additional centromeric plasmid or artificial chromosome. Boxes represent means and standard deviation. Means were compared using a Student's t-test. Red circles indicate centromeres. The new data was also added to the summary of epistatic effects (Figure 4f):

New Figure 4f | Summary of epistatic effects of different perturbations. Both diploidization, adding a centromeric plasmid, and an artificial chromosome increase the inward force, and adding benomyl or deleting KIP1 decrease the outward force. Boxes represent means and standard deviation.

Additionally, we constructed diploid strains with tagged SPBs and quantified the mitotic defect in strains with $n = 16, 8, 3,$ and 2 chromosomes to show that both the growth rescue we observed for the diploid 3 chromosome strain and the growth defect of the diploid 2 chromosome strain (Fig. 3b) are indeed a mitotic phenotype (new Fig. 3e and new Extended Data Figure 3). Together, these observations indicate that it is not the number of chromosomes (plasmid and artificial chromosome have the same effect), the amount of DNA (diploids have double the amount of DNA and the cut-off is still 4 CENs total), but really the total number of CENs in the cell that determines whether the cell has a growth/mitotic defect.

*New Figure 3e | The time from SPB doubling to max. anaphase separation for diploid strains. Boxes represent the means and standard deviation. Means were compared using a Student's t-test; * $p < 0.05$.*

New Extended Data Figure 3 | Distance between SPBs over time for diploidized strains. For normalization, the time point with maximal SPB separation during anaphase was set to zero.

Finally, we also consider that even though the growth effect in Fig 3a is subtle, we observe the exact same rescuing effect in our independent microscopy experiments (Fig. 3c and 3d), both in terms of mitotic timing (Fig. 3d), as well as the spindle elongation phenotype during metaphase (Fig. 4g). We remade Fig. 4b and Fig. 4g to more clearly highlight and explain this phenotype: we changed the graphs to show the distance between SPBs at the end of metaphase instead to make it easier to compare between the different strains and treatments.

New Figure 4b and 4g | (b) Distance between SPBs at the end of metaphase for different fusion strains. Boxes represent means and standard deviation. (g) Distance between SPBs at the end of metaphase for different fusion strains and the effects of increasing inward force (+ pCEN) or decreasing outward force (+ benomyl). Means were compared using a Student's t-test; * $p < 0.05$, *** $p < 0.001$.

2. The claim on “the force imbalance causes kinetochore declustering and triggers the SAC”, seems not fully supported. Statistically analysis of the declustering is required. On the other hand, besides using the Mad2 mutant, additional evidence of activation of SAC is needed.

Our manuscript now contains a quantitative analysis of the declustering phenotype (new Figure 5e). Using new snapshots of exponentially growing cultures, we determined the proportion of cells with a bilobed kinetochore signal (i.e. bimodal Ndc80 signal along the spindle pole to spindle pole axis) in both metaphase and anaphase. For the strain with three chromosomes, we observe significantly fewer cells with a bilobed kinetochore distribution during metaphase, but this difference disappears during anaphase (see also the representative time lapses shown in the same figure). By adding benomyl, the proportion of cells with a bilobed kinetochore signal can be restored to wild-type levels.

New Figure 5e | Proportion of cells with a bilobed kinetochore signal (i.e. bimodal Ndc80 signal along the spindle pole to spindle pole axis) in both metaphase and anaphase. Proportions were compared using a two-proportions Z-test; * $p < 0.05$.

To provide additional evidence of SAC activation, we followed *Pds1* (securin) levels upon release from a G1 arrest. During metaphase, securin binds and inhibits securase, the protease that degrades cohesin. During the metaphase-to-anaphase transition, securin is degraded, releasing separase so cohesin can be degraded and sister chromatids unlinked. SAC activation stabilises *Pds1*, and therefore prevents this transition. We switched the mating type of our strains (to allow for G1 arrest with alpha factor), and tagged *Pds1* with a 3xHA tag. After release from G1 arrest, we collected samples at specific timepoints, and performed a Western blot to quantify *Pds1* levels (new Figure 5d and new Extended Data Figure 5f).

New Figure 5d | Western blot analysis of *Pds1* levels after G1 release for 16- and 3-chromosome strains. Cells were collected at the indicated time points and alpha factor was added again 45 minutes after release to prevent the cells from entering a second cell cycle. Actin levels were used as loading control. *Pds1*-normalised values are shown in the bar plot at the bottom.

New Extended Data Figure 5f | Western blot analysis of *Pds1* levels after G1 release for 16- and 3-chromosome strains, focussed on the second cell cycle after release. Cells were collected at the indicated time points. Ponceau S staining was used as loading control. *Pds1*-normalised values are shown in the bar plot at the bottom.

Consistent with both our population growth experiments and single cell microscopy - both showing a 5-10 minute delay in growth rate and duration of mitosis respectively - the 3-chromosome strain

displays higher *Pds1* levels at the end of the cell cycle compared to the wild type, indicating that metaphase and SAC activation persist longer in a significant proportion of the population. As expected for a 5-10 minute delay, the difference is small (~15% higher *Pds1* levels in 3-chromosome strain at the relevant time points), but consistent across independent experiments and can be observed in both the first and second cell cycle after G1 release.

3. Besides the spindle segregation phenotype, are there any other potential causes for the growth defects such as activation of cell cycle checkpoint etc. An survey on these potentials at the begin will enhance the quality of this manuscript

Apart from exploring the effect of deleting a component of the spindle assembly checkpoint, we now also test whether deleting a component of the DNA damage checkpoint (*RAD9Δ*), increasing DNA damage (+hydroxyurea), or deleting a component of the spindle positioning checkpoint (*BUB2Δ*) influence the defect in the 3-chromosome strain (new Extended Data Figure 5b-d). None of these perturbations alleviate (or exacerbate) the defect.

New Extended Data Figure 5b-d | (b) Maximum growth rates of fused-chromosome strains with and without *RAD9* deletion. Boxes represent means and standard deviations. Means were compared using a Student's *t*-test; ** $p < 0.01$. (c) Maximum growth rates of fused-chromosome strains with and without 10 mM hydroxyurea. Boxes represent means and standard deviations. Means were compared using a Student's *t*-test; *** $p < 0.001$. (d) Maximum growth rates of fused-chromosome strains with and without *BUB2* deletion. Boxes represent means and standard deviations. Means were compared using a Student's *t*-test; *** $p < 0.001$.

Minor concerns:

1. Did you try the strain with only 1 or 2 chromosomes? I am curious how these strains behave in the analysis. It is still a mystery why the 2 chromosomes can't be fused into 1 in Luo's paper. (This is not required for the publication of this manuscript)

We used the 2-chromosome strain for growth rate measurements (both haploid and diploid), and now we have additional microscopic evidence of the mitotic defect in the 2-chromosome diploid (see above).

We first planned to also use the single-chromosome strain in our analyses, but when handling the strain it quickly became apparent that it is more unstable than the other strains and not suited for

our type of analyses: unevolved, log-phase populations that are started from our frozen stock already show extensive variability in cell size:

Scale = 10 μ m

2. To rule out potential effect from double the gene content, overall transcription analysis might be useful in both haploid and diploid strains in Fig2

It would indeed be interesting to look at the transcriptome of the haploids, diploids, and evolved strains. However, we believe that with the addition of the additional data shown above (response to comment 1), we now have sufficient evidence to confidently support our hypothesis that increasing the number of centromeres explains why diploidization is adaptive in the 3-chromosome strain.

3. Fig4b/4g, color is too close. It's hard to distinguish different samples

We changed the graphs to show the distance between SPBs at the end of metaphase instead to make it easier to compare between the different strains and treatments. See response to comment 1 above.

4. Adding analysis of epistatic effects between #of chromosome and the SAC mutants, similar to Fig4f

A graph with the epistatic effects was added to the supplements (Extended data figure 5a).

New Extended Data Figure 5a | Epistatic effect of MAD2 deletion. Boxes represent means and standard deviation.

5. Fig5c, adding p-value or *

Statistical values have been added.

Decision Letter, first revision:

Our ref: NCB-A52680A

17th June 2024

Dear Dr. Dey,

Thank you for submitting your revised manuscript "Spindle architecture constrains karyotype in budding yeast" (NCB-A52680A). It has now been seen by the original Referees #1 and #3 and their comments are below. The reviewers find that the paper has improved in revision, and therefore we'll be happy in principle to publish it in Nature Cell Biology, pending minor revisions to comply with our editorial and formatting guidelines.

The current version of your manuscript is in a PDF format; could you please email us a copy of the file in an editable format (Microsoft Word or LaTeX), as we can not proceed with PDFs at this stage? Thank you very much in advance for your attention to this point.

Once we have the Word file, we will begin performing detailed checks on your paper and will send you a checklist detailing our editorial and formatting requirements within about 2 weeks. Please do not upload the final materials and make any revisions until you receive this additional information from us.

Thank you again for your interest in Nature Cell Biology. Please do not hesitate to contact me if you have any questions.

Sincerely,

Melina

Melina Casadio, PhD
Senior Editor, Nature Cell Biology
ORCID ID: <https://orcid.org/0000-0003-2389-2243>

Reviewer #1 (Remarks to the Author):

The authors have satisfactorily addressed all our concerns. In particular, the clarification of the number of centromeres (not chromosomes) as the rate-limiting variable during rapid karyotype evolution now makes the paper stronger and with a clear and important message to this rapidly emerging field of evolutionary cell biology. The authors are to be congratulated for this very elegant work.

Helder Maiato and Inês Dias

Reviewer #3 (Remarks to the Author):

All my concerns have been adequately addressed

Decision Letter, final checks:

Our ref: NCB-A52680A

25th June 2024

Dear Dr. Dey,

Thank you for your patience as we've prepared the guidelines for final submission of your Nature Cell Biology manuscript, "Spindle architecture constrains karyotype in budding yeast" (NCB-A52680A). Please carefully follow the step-by-step instructions provided in the attached file, and add a response in each row of the table to indicate the changes that you have made. Ensuring that each point is addressed will help to ensure that your revised manuscript can be swiftly handed over to our production team.

In recognition of the time and expertise our reviewers provide to Nature Cell Biology's editorial process, we would like to formally acknowledge their contribution to the external peer review of your manuscript entitled "Spindle architecture constrains karyotype in budding yeast". For those reviewers who give their assent, we will be publishing their names alongside the published article.

Nature Cell Biology offers a Transparent Peer Review option for new original research manuscripts submitted after December 1st, 2019. As part of this initiative, we encourage our authors to support increased transparency into the peer review process by agreeing to have the reviewer comments, author rebuttal letters, and editorial decision letters published as a Supplementary item. When you submit your final files please clearly state in your cover letter whether or not you would like to participate in this initiative. Please note that failure to state your preference will result in delays in accepting your manuscript for publication.

Cover suggestions

COVER ARTWORK: We welcome submissions of artwork for consideration for our cover. For more

information, please see our guide for cover artwork.

Nature Cell Biology has now transitioned to a unified Rights Collection system which will allow our Author Services team to quickly and easily collect the rights and permissions required to publish your work. Approximately 10 days after your paper is formally accepted, you will receive an email in providing you with a link to complete the grant of rights. If your paper is eligible for Open Access, our Author Services team will also be in touch regarding any additional information that may be required to arrange payment for your article.

Please note that *Nature Cell Biology* is a Transformative Journal (TJ). Authors may publish their research with us through the traditional subscription access route or make their paper immediately open access through payment of an article-processing charge (APC). Authors will not be required to make a final decision about access to their article until it has been accepted. Find out more about Transformative Journals

Please use the following link for uploading these materials:
[Redacted]

Best regards,

Kendra Donahue
Staff
Nature Cell Biology

On behalf of

Melina Casadio, PhD
Senior Editor, Nature Cell Biology
ORCID ID: <https://orcid.org/0000-0003-2389-2243>

Reviewer #1:

Remarks to the Author:

The authors have satisfactorily addressed all our concerns. In particular, the clarification of the number of centromeres (not chromosomes) as the rate-limiting variable during rapid karyotype evolution now makes the paper stronger and with a clear and important message to this rapidly emerging field of evolutionary cell biology. The authors are to be congratulated for this very elegant work.

Helder Maiato and Inês Dias

Reviewer #3:

Remarks to the Author:

All my concerns have been adequately addressed

Final Decision Letter:

Dear Dr Dey,

I am pleased to inform you that your manuscript, "Spindle architecture constrains karyotype evolution", has now been accepted for publication in Nature Cell Biology. Congratulations on this beautiful study!

Due to the importance of these deadlines, we ask that you please let us know now whether you will be difficult to contact over the next month. If this is the case, we ask you provide us with the contact

information (email, phone and fax) of someone who will be able to check the proofs on your behalf, and who will be available to address any last-minute problems.

Please note that *Nature Cell Biology* is a Transformative Journal (TJ). Authors may publish their research with us through the traditional subscription access route or make their paper immediately open access through payment of an article-processing charge (APC). Authors will not be required to make a final decision about access to their article until it has been accepted. Find out more about Transformative Journals

If you have not already done so, we strongly recommend that you upload the step-by-step protocols

used in this manuscript to protocols.io (<https://protocols.io>), an open online resource that allows researchers to share their detailed experimental know-how. All uploaded protocols are made freely available and are assigned DOIs for ease of citation. Protocols and Nature Portfolio journal papers in which they are used can be linked to one another, and this link is clearly and prominently visible in the online versions of both. Authors who performed the specific experiments can act as primary authors for the Protocol as they will be best placed to share the methodology details, but the Corresponding Author of the present research paper should be included as one of the authors. By uploading your Protocols onto protocols.io, you are enabling researchers to more readily reproduce or adapt the methodology you use, as well as increasing the visibility of your protocols and papers. You can also establish a dedicated workspace to collect your lab Protocols. Further information can be found at <https://www.protocols.io/help/publish-articles>.

With kind regards,

Melina Casadio, PhD
Senior Editor, Nature Cell Biology
ORCID ID: <https://orcid.org/0000-0003-2389-2243>